# Extremely low-frequency pulses of faint magnetic field induce mitophagy to rejuvenate mitochondria

Takuro Toda[1], Mikako Ito [1], Jun-ichi Takeda [1], Akio Masuda[1], Hiroyuki Mino[2], Nobutaka Hattori[3], Kaneo Mohri[4] & Kinji Ohno [1✉]

Humans are frequently exposed to time-varying and static weak magnetic fields (WMF). However, the effects of faint magnetic fields, weaker than the geomagnetic field, have been scarcely reported. Here we show that extremely low-frequency (ELF)-WMF, comprised of serial pulses of 10 μT intensity at 1–8 Hz, which is three or more times weaker than the geomagnetic field, reduces mitochondrial mass to 70% and the mitochondrial electron transport chain (ETC) complex II activity to 88%. Chemical inhibition of electron flux through the mitochondrial ETC complex II nullifies the effect of ELF-WMF. Suppression of ETC complex II subsequently induces mitophagy by translocating parkin and PINK1 to the mitochondria and by recruiting LC3-II. Thereafter, mitophagy induces PGC-1α-mediated mitochondrial biogenesis to rejuvenate mitochondria. The lack of PINK1 negates the effect of ELF-WMF. Thus, ELF-WMF may be applicable for the treatment of human diseases that exhibit compromised mitochondrial homeostasis, such as Parkinson's disease.

[1] Division of Neurogenetics, Center for Neurological Diseases and Cancer, Nagoya University Graduate School of Medicine, Nagoya, Japan. [2] Division of Material Science, Nagoya University Graduate School of Science, Nagoya, Japan. [3] Department of Neurology, Juntendo University, Tokyo, Japan. [4] Nagoya Industrial Science Research Institute, Nagoya, Japan. ✉email: ohnok@med.nagoya-u.ac.jp

In the present-day industrialized societies, humans are exposed daily to time-varying and static weak magnetic fields (WMF). The effects of WMF on animals, including humans, have been documented in a few reports. WMF increases intracellular calcium concentrations and induces the development of satellite cells[1]. Similarly, static magnetic fields increase cytosolic calcium and reactive oxygen species (ROS) in mouse embryonic stem (ES) cell-derived embryoid bodies and Flk-1+ cardiac progenitor cells[2], although the magnetic intensities were as much as 0.3–5.0 mT. In addition, static WMF as weak as 0.01 μT reduces the ROS level in nonactivated neutrophils[3]. Moreover, the exposure to static WMF of 200–600 μT in HT1080 cells increased the mitochondrial calcium concentration and the mitochondrial membrane potential[4]. For a disease model, the viability of breast cancer cells is specifically decreased by WMF[5]. Extremely low-frequency WMF (ELF-WMF), which is defined as ELF with a frequency of 300 Hz or less, may or may not reduce the levels of ROS in cells[6]. ROS are mostly produced during electron transfer through the mitochondrial electron transport chain (ETC). The biological effects of ELF-WMF, weaker than the geomagnetic field, have been reported in cultured cells[7], planaria[8], rats[9], lizards[10,11], and humans[12], but the underlying mechanisms remain elusive. In addition, the optimal conditions for the manifestation of the cellular effects of ELF-WMF remain undetermined. Furthermore, the molecular mechanisms underlying the effects of ELF-WMF have not been elucidated.

Mitophagy and mitochondrial biogenesis cooperate in the maintenance of mitochondrial homeostasis. Mitophagy is a quality-assurance system that selectively eliminates damaged mitochondria using the macroautophagy machinery. Mitophagy-associated proteins include PTEN-induced kinase 1 (PINK1), parkin, BCL2/adenovirus E1B 19 kDa protein-interacting protein 3 (BNIP3), BCL2/adenovirus E1B 19 kDa protein-interacting protein 3-like (NIX/BNIP3L), and FUN14 domain containing 1 (FUNDC1). Mitophagy is induced by mitochondrial damage, excessive levels of mitochondrial ROS, endoplasmic reticulum (ER) stress[13], circadian rhythm[14], and hypoxia-inducible factor[15]. In addition, mitophagy induces mitochondrial biogenesis to compensate for the removal of mitochondria[16,17]. The parkin/PINK1 pathway is a key regulator of mitophagy. Parkin and PINK1 translocate from the cytosol to the mitochondria, and ubiquitinate the mitochondria, which are subsequently recognized by the phagosome-lining LC3 to eliminate the mitochondria. Accumulating knowledge points to the notion that the compromised parkin/PINK1 pathway is associated with the development and progression of neurodegenerative diseases, including Parkinson's disease[18] and Alzheimer's disease[19].

In this study, we investigated the effects of faint magnetic fields, weaker than the geomagnetic field, on cells. We report that ELF-WMF efficiently suppresses the mitochondrial mass to 70% by inhibiting the mitochondrial ETC complex II, which subsequently induces mitophagy and rejuvenates mitochondria. We expect that ELF-WMF may be applicable to a plethora of human diseases that exhibit compromised mitophagy like neurodegenerative diseases.

## Results

### Exposure to Opti-ELF-WMF for 4 weeks increases the mitochondrial activity in the mouse liver. Based on our previous observation that 1–8 Hz stimulation of 4 ms pulses of 10 μT magnetic field (Opti-ELF-WMF) decreases the thermal hysteresis of electric resistance of modified Ringer's solution[20,21], we examined the effect of 4-week exposure to Opti-ELF-WMF on the liver mitochondria in wild-type C57BL6/N mice. Using an open-field locomotor test, we first confirmed that exposure to Opti-

ELF-WMF had no effect on the locomotor activity in mice (Supplementary Fig. 1e, f). We examined the ETC activity of mitochondria isolated from the mouse liver by measuring the oxygen consumption rate (OCR) and mitochondrial membrane potential using a flux analyzer and tetramethylrhodamine (TMRM), respectively. We found that Opti-ELF-WMF increased both the OCR and mitochondrial membrane potential by approximately 40% (Fig. 1a, b).

Next, we evaluated the enzymatic activity of each ETC complex and the amount of OXPHOS proteins in the mouse liver homogenates. Opti-ELF-WMF increased the activities of mitochondrial ETC complexes I to IV, although statistical significance was observed only in complex IV (Fig. 1c). The levels of four nucleus-encoded proteins (NDUFB8 [complex I], SDHB [complex II], UQCRC2 [complex III], and ATP5F1A [complex V]) also tended to be increased (Fig. 1d). In contrast, the levels of mitochondria-encoded MTCO1 [complex IV] remained unchanged.

These data demonstrate that Opti-ELF-WMF had no effect on the locomotor activity in wild-type mice, but tended to increase the mitochondrial ETC complex activities and the levels of nucleus-encoded ETC proteins in the mouse liver.

### Opti-ELF-WMF temporarily decreases the mitochondrial ROS levels, mitochondrial mass, and mitochondrial membrane potential in cultured cells. To further dissect the effect of Opti-ELF-WMF on the mitochondria, mouse hepatocyte-derived AML12 cells were cultured under Opti-ELF-WMF for 1 to 24 h, and were stained with MitoSOX, MitoTracker Green, and TMRM to quantify the levels of mitochondrial superoxide, mitochondrial mass, and mitochondrial membrane potential, respectively. Opti-ELF-WMF most strongly decreased the level of mitochondrial superoxide at 1 h, mitochondrial mass at 3 h, and mitochondrial membrane potential at 6 h, and most strongly increased them at 12 h (Fig. 2a–c). At 24 h, the values reverted to normal levels. Thus, Opti-ELF-WMF suppressed the mitochondrial ETC activity at 1 h, which was likely to be followed by elimination and/or inactivation of a subset of mitochondria at 3 to 6 h. The mass and function of mitochondria were then increased at 12 h and returned to normal levels at 24 h. These time points were used for subsequent analyses.

### Optimal conditions of ELF-WMF for the reduction of the mitochondrial mass in cultured cells. Next, we analyzed the optimal conditions of ELF-WMF by measuring the mitochondrial mass at 3 h by changing the intensity, pulse width, and frequency of ELF-WMF. ELF-WMF, less than 10 μT, showed MF intensity-dependent reduction in the mitochondrial mass, but the effects were not enhanced when the MF intensity ranged from 10 to 200 μT (Supplementary Fig. 2a). However, compared to 10 μT, 300 μT ELF-WMF had a marginally reduced effect. ELF-WMF with pulse widths of 2, 4, and 8 ms reduced the mitochondrial mass, with a peak at 4 ms (Supplementary Fig. 2b). In contrast, ELF-WMF with pulse widths of 1 and 16 ms had no effect. The mitochondrial mass was reduced the most upon treatment with increasing frequencies of ELF-WMF (1, 2, 3, 4, 5, 6, 7, and 8 Hz for 1 s each) (Supplementary Fig. 2c). Static frequencies at 6 and 8 Hz had no effect. Similarly, changing the frequency profiles to 1–4 Hz for 1 s each or to 1–16 Hz for 1 s each had no effect.

As observed for AML12 cells (Fig. 2b, c), Opti-ELF-WMF first reduced the mitochondrial mass, and thereafter increased the mitochondrial membrane potential in Neuro2A, C2C12, human iPS, HEK293, and HeLa cells (Supplementary Table 2). Thus, the effects of Opti-ELF-WMF on mitochondria are unlikely to be cell line-specific.

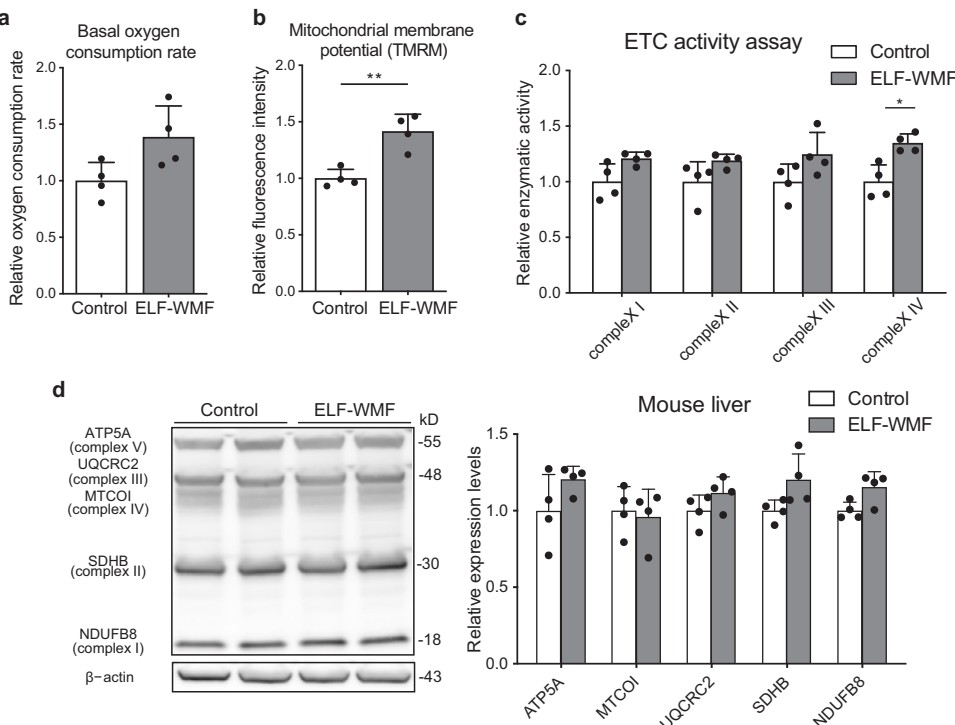

**Fig. 1 Exposure to Opti-ELF-WMF for 4 weeks increased mitochondrial electron transport chain (ETC) activities in the mouse liver. a** Basal oxygen consumption rate was measured by a flux analyzer using mitochondria isolated from the mouse liver. No statistically significant difference was observed by Student's *t*-test (mean ± SD, *n* = 4 mice each). **b** Membrane potential of mitochondria isolated from the mouse liver was measured by flow cytometry with tetramethylrhodamine (TMRM; mean ± SD, *n* = 4 mice each; **p < 0.01 by Student's *t*-test). **c** Relative enzymatic activities of mitochondrial electron transport chain (ETC) complexes I, II, III, and IV of the mouse liver (mean ± SD, *n* = 4 mice each; *q [false discovery rate] < 0.05 by multiple Student's *t*-tests). **d** Western blotting of the mitochondrial oxidative phosphorylation proteins in the mouse liver (mean ± SD *n* = 4 mice each; no statistical difference was observed by false discovery rate with multiple Student's *t*-tests). Representative duplicates of Western blot analysis are shown. See also Fig. S1.

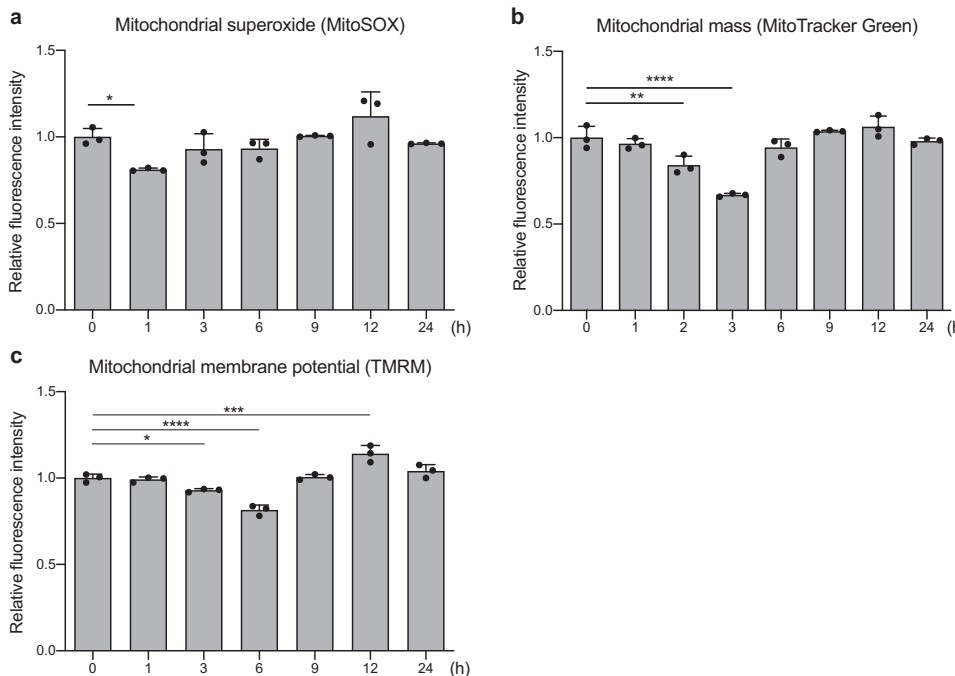

**Fig. 2 Opti-ELF-WMF temporarily decreased the levels of mitochondrial reactive oxygen species (ROS), mitochondrial mass, and mitochondrial membrane potential. a** Mitochondrial ROS of mouse hepatocyte-derived AML12 cells exposed to Opti-ELF-WMF for 1 to 24 h was evaluated by MitoSOX. **b** Mitochondrial mass of AML12 cells exposed to Opti-ELF-WMF for 1 to 24 h was evaluated by MitoTracker Green. **c** Mitochondrial membrane potential of AML12 cells exposed to Opti-ELF-WMF for 1 to 24 h was evaluated by tetramethylrhodamine (TMRM). **a–c** Show mean ± SD, *n* = 3 culture dishes each; *p < 0.05, **p < 0.01, ***p < 0.001, and ****p < 0.0001 by one-way ANOVA followed by Dunnett's post hoc test compared with the value at time 0.

**Decrease in the mitochondrial mass by Opti-ELF-WMF is accounted for by temporary decreases in the levels of mitochondrial ETC proteins and of outer membrane proteins.** To identify the mitochondrial proteins that were decreased by the Opti-ELF-WMF exposure, we quantified the amounts of mitochondrial ETC proteins and VDAC1, which is an outer membrane protein, by Western blot analysis. Opti-ELF-WMF had no effect on the levels of the eight examined mitochondrial proteins at 1 h (Supplementary Fig. 3), but decreased them at 3 h (Fig. 3). At 12 h, the amounts of five proteins (NDUFB8, SDHB, UQCRC2, MTCO1, and VDAC1) were restored to their basal levels, and those of three proteins (ATP5A, NDUFS1, and UQCRFS1) were increased compared to their basal levels. Taken together, Opti-ELF-WMF decreased the levels of all the examined mitochondrial proteins at 3 h, which was consistent with the decreased mitochondrial mass at 3 h after exposure (Fig. 2b). The levels of mitochondrial proteins were restored to normal or higher than normal levels at 12 h.

**Opti-ELF-WMF induces mitophagy.** Mitophagy is an autophagic mechanism in the mitochondrial quality assurance system that eliminates damaged mitochondria. We investigated whether mitophagy is activated by Opti-ELF-WMF. We first examined the expression levels of mitophagy-related proteins, PINK1 and LC3-II, in whole cell lysates of AML12 cells. PINK1 triggers mitophagy, whereas LC3-II is an effector that eliminates the mitochondria. The amount of PINK1 gradually increased until 90 min and gradually decreased thereafter upon exposure to Opti-ELF-WMF (Fig. 4a). Similarly, the amount of LC3-II gradually increased until 120 min and gradually decreased thereafter (Fig. 4a). These results indicated that the decrease in mitochondrial mass by Opti-ELF-WMF was likely due to the activation of mitophagy.

Next, we isolated the mitochondrial and cytosolic fractions of AML12 cells, and examined the purity by immunoblotting of β-actin and VDAC1, respectively (Fig. 4b). We evaluated the expression levels of PINK1 and parkin, and mitochondrial ubiquitination at 120 min in the mitochondrial and cytosolic fractions. PINK1 and parkin translocate from the cytosol to the mitochondria, and ubiquitinate mitochondrial proteins. Opti-ELF-WMF increased the level of parkin in the mitochondria, but had no effect in the whole cells (Fig. 4b). In contrast, Opti-ELF-WMF increased the levels of PINK1 in both mitochondria and whole cells (Fig. 4b). We also found that Opti-ELW-WMF induced the ubiquitination of mitochondrial proteins (Fig. 4b). These results indicated that Opti-ELF-WMF accumulated PINK1 and parkin in the mitochondria, and induced mitochondrial ubiquitination. To detect the mitochondria in the lysosomes, we used the Mtphagy Dye that fluoresces with decreasing pH around mitochondria. The fluorescence intensity of the Mtphagy Dye peaked at 150 min (Fig. 4c). We also confirmed the colocalization of mitochondria and lysosomes at 150 min by confocal microscopy (Fig. 4d).

To further confirm the effect of PINK1 on Opti-ELF-WMF-induced mitophagy, we examined the mitochondrial mass and the amounts of ATP5A and VDAC1 in PINK1-knocked out (KO) HeLa cells[22]. As expected, exposure of PINK1-KO HeLa cells to Opti-ELF-WMF failed to reduce the mitochondrial mass (Supplementary Fig. 4a), or the levels of mitochondrial proteins (Supplementary Fig. 4b).

**PGC-1α expression is upregulated for mitochondrial biogenesis after mitophagy.** PGC-1α is a key player in mitochondrial biogenesis. PPARα and TFAM are regulated by PGC-1α, and are effectors of mitochondrial biogenesis and metabolism. Thus, we examined whether the recovery of mitochondrial mass was mediated by PGC-1α, TFAM, and PPARα. We observed that Opti-ELF-WMF increased the expression of these proteins at 12 h (Fig. 5), indicating that PGC-1α-mediated mitochondrial biogenesis was activated after mitophagy to rejuvenate mitochondria.

**Opti-ELF-WMF suppresses the enzymatic activity of ETC complex II in vitro.** We conducted RNA-sequence (RNA-seq) analysis along with gene set enrichment analysis (GSEA) using AML12 cells exposed to Opti-ELF-WMF for 1 h. We found that Opti-ELF-WMF reduced the expression of mitochondrial ETC genes (Table 1). Thus, suppression of mitochondrial ETC genes is likely to be a key factor in the triggering of mitophagy by Opti-ELF-WMF.

To capture the initial event activated by Opti-ELF-WMF, we examined the direct effects of Opti-ELF-WMF on the enzymatic activities of mitochondrial ETC complexes I, II, III, and IV in mouse liver homogenates that were exposed to Opti-ELF-WMF for 8 min in vitro. The enzymatic activity of ETC complex II was reduced to 88% by Opti-ELF-WMF, whereas the activities of the other ETC complexes (I, III, and IV) remained unchanged (Fig. 6a). Mitochondrial ETC complex II is comprised of four succinate dehydrogenase (SDH) subunits: SDHA, SDHB, SDHC, and SDHD (Fig. 6e). To further dissect the effect of Opti-ELF-WMF on ETC complex II, we quantified the enzymatic activities of succinate:quinone reductase (SQR), succinate cytochrome c reductase (SCR), SDH, and SDHA in the mitochondria isolated from mouse liver homogenates exposed to Opti-ELF-WMF for 8 min. The activities of SQR, SCR, SDH, and SDHA decreased to 85%, 85%, 90%, and 95%, respectively (Fig. 6b). Thus, ELF was likely to suppress all the four subunits of mitochondrial ETC complex II. We also examined the expression of SDHA, SDHB, SDHC, and SDHD under Opti-ELF-WMF and did not find any change in the expression levels of these subunits at 1 h after exposure (Supplementary Fig. 5).

To examine whether mitophagy by Opti-ELF-WMF was indeed due to the suppression of mitochondrial ETC complex II, AML12 cells were incubated with either 3-nitropropionic acid (3-NP), an ETC complex II inhibitor, or rotenone, an ETC complex I inhibitor, for 12 h. The cells were then exposed to Opti-ELF-WMF for 3 h. Inhibition of ETC complex II by 3-NP negated the reduction in the mitochondrial mass induced by Opti-ELF-WMF (Fig. 6c). In contrast, inhibition of ETC complex I had no effect on the reduction in the mitochondrial mass induced by Opti-ELF-WMF (Fig. 6d). Taken together, Opti-ELF-WMF-mediated mitophagy requires electron flow through mitochondrial ETC complex II.

## Discussion
We found that Opti-ELF-WMF reduced the amount of mitochondria by ~30% (Fig. 2b) by inhibiting mitochondrial ETC complex II by ~15% (Fig. 6a). This subsequently induced mitophagy (Fig. 4) to eliminate damaged mitochondria, and later activated mitochondrial biogenesis (Fig. 5) to increase mitochondrial membrane potential (Fig. 2c). To examine the long-term effects of Opti-ELF-WMF, we exposed wild-type mice to Opti-ELF-WMF for 4 weeks, and observed increased mitochondrial membrane potential in the mouse liver by ~40% (Fig. 1b). Mitochondrial ETC complex II is comprised of four subunits, and Opti-ELF-WMF suppressed the activities of all the four subunits. The optimal conditions for ELF-WMF exposure to suppress the mitochondrial ETC activities were 1–8 Hz serial pulses for every 1 s, 10 μT magnetic field, and 4 ms pulse width, which are referred to as Opti-ELF-WMF in this communication. We previously reported that Opti-ELF-WMF most efficiently decreased the

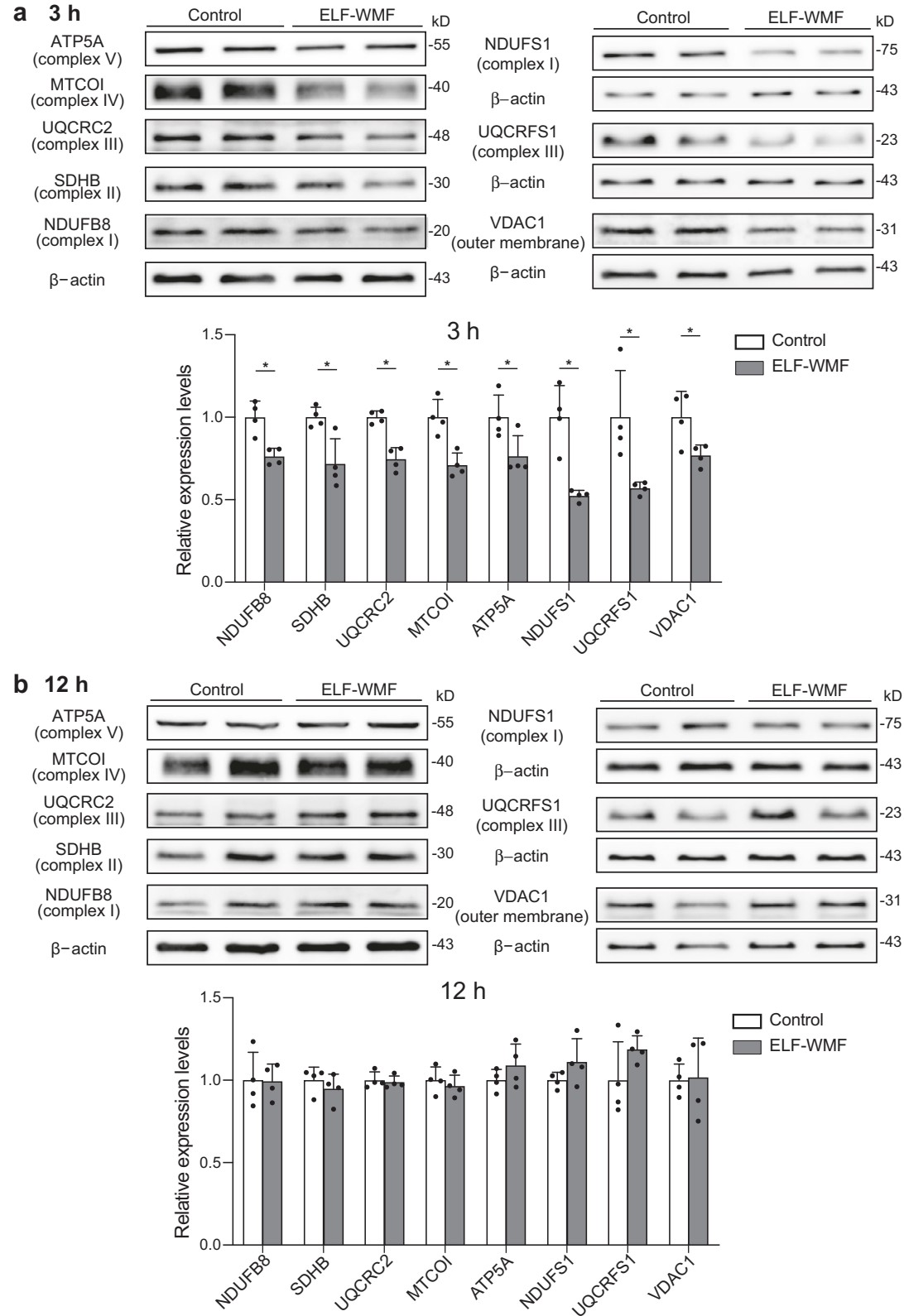

**Fig. 3 Temporary downregulation of the levels of mitochondrial proteins at 3 h and recovery or slight increase at 12 h in cells exposed to Opti-ELF-WMF.** Representative duplicates of Western blot analysis of mitochondrial oxidative phosphorylation proteins, as well as of a mitochondrial outer membrane protein, VDAC1, in AML12 cells are shown at 3 h (**a**) and 12 h (**b**) exposure to Opti-ELF-WMF (mean ± SD, $n = 4$ culture dishes each; *$q < 0.05$, and **$q < 0.01$ by multiple Student's $t$-tests). See also Fig. S3.

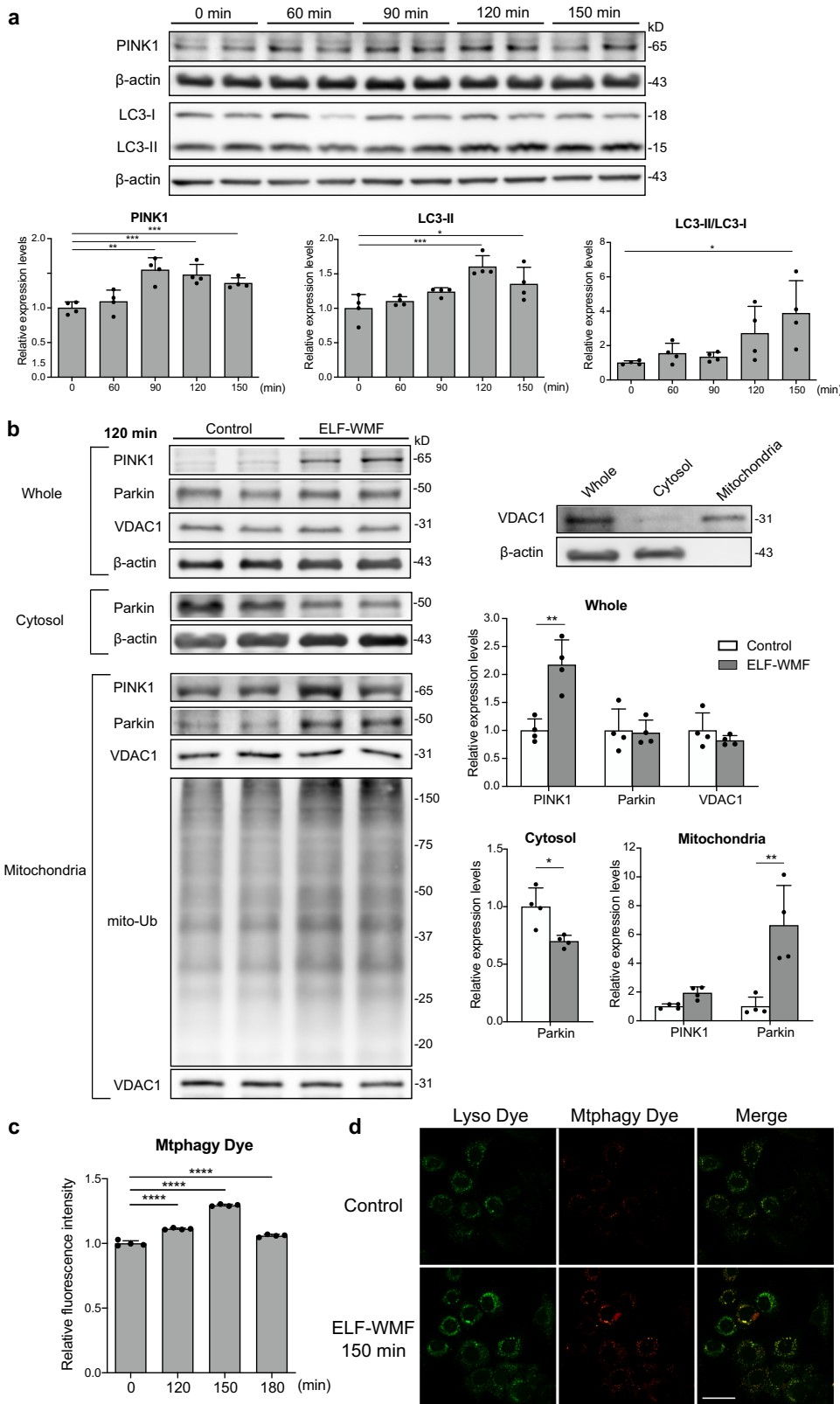

hysteresis of the electronic resistance of modified Ringer's solution as a function of temperature[20,21]. We have shown the effects of Opti-ELF-WMF on multiple cultured cells and wild-type mice for the first time. We found that the conditions of Opti-ELF-WMF exhibited the maximum effect on the reduction in the mitochondrial mass (Supplementary Fig. 2). The modifications of the pulse widths (Supplementary Fig. 2b) and the frequency profiles (Supplementary Fig. 2c) markedly attenuated the effects of ELF-WMF on the mitochondrial mass. As the conditions of Opti-ELF-WMF exerted the maximum effects on the hysteresis of electronic resistance of modified Ringer's solution in vitro and on the reduction of mitochondrial mass in vivo, the identity of the molecular target of Opti-ELF-WMF on ETC complex II subunits may share a feature similar to that of modified Ringer's solution.

**Fig. 4 Opti-ELF-WMF induced mitophagy. a** AML12 cells were exposed to Opti-ELF-MF for up to 150 min. The levels of PINK1 and LC3-II were evaluated by Western blot analysis (mean ± SD, $n = 4$ culture dishes each; *$p < 0.05$, **$p < 0.01$, and ***$p < 0.001$ by one-way ANOVA followed by Dunnett's post hoc test compared with the value at time 0). **b** Western blot analysis of β-actin and VDAC1 in whole, cytosolic, and mitochondrial fractions of AML12 cells to indicate the purity of each fraction. Representative duplicated Western blot analysis for parkin and PINK1 in whole cell lysates, a cytosolic fraction, and a mitochondrial fraction, as well as for ubiquitination in a mitochondrial fraction of AML12 cells exposed to Opti-ELF-WMF for 120 min (mean ± SD, $n = 4$ culture dishes each; *$q < 0.05$ and **$q < 0.01$ by multiple Student's $t$-tests). **c** Mitophagy was evaluated using Mtphagy Dye in AML12 cells exposed to Opti-ELF-WMF for up to 180 min (mean ± SD, $n = 4$ culture dishes each; ****$p < 0.0001$ by one-way ANOVA followed by Dunnett's posthoc test compared with the value at time 0). **d** Representative confocal images showing colocalization of mitochondria, detected by Mtphagy Dye, and lysosomes, detected by Lyso Dye, in AML12 cells exposed to Opti-ELF-WMF for 150 min. Scale bar, 50 μm. See also Fig. S4.

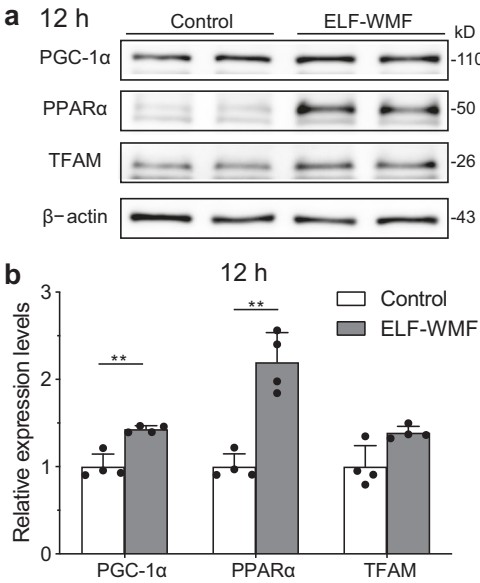

**Fig. 5 Opti-ELF-WMF induced mitochondrial biogenesis. a** Representative duplicates of Western blot analysis are shown. **b** Densitometric analysis of Western blots (mean ± SD, $n = 4$ culture dishes each; *$q < 0.05$ and **$q < 0.01$ by multiple Student's $t$-tests).

The effects of magnetic fields on cultured cells, animal models, and humans have been reported mostly using static magnetic fields (SMFs) and radio frequency magnetic fields (RF-MF)[6,23,24]. Similarly, the effects of ELF-MF with static frequencies have also been reported[6,25], but the effect of ELF-MF with time-varying frequencies has not been reported. In addition, the intensities of SMF, RF-MF, and ELF-MF were mostly greater than 1 mT and were rarely less than 100 μT. In contrast, Opti-ELF-WMF had an MF intensity of 10 μT. According to the guidelines for limiting exposure to time-varying electric and magnetic fields by the International Commission on Non-Ionizing Radiation Protection (ICNIRP)[26], the intensities of time-varying MF acceptable for occupational exposure increase with decreasing frequencies. For example, MF intensities of 1 mT and lower are safe at less than 300 Hz. Adverse effects with higher MF intensities include induction of magnetic phosphenes by 5 mT ELF-MFs at 20 Hz[27]; gross external, visceral, or skeletal malformations by 20 mT LF-MF[28,29]; and genotoxicity to cells by 50 mT LF-MF[30]. Because beneficial biological effects are sometimes inevitably accompanied by adverse effects, it is reasonable that biological effects of all the modalities of MF have been studied mostly with 1 mT or higher intensities.

Previous studies have shown that SMF, RF-MF, and ELF-MF increase, decrease, or have no effect on the levels of ROS in cultured cells and animal models[6]. In an SMF study, 200 μT decreased and 500 μT increased a surrogate marker of ROS, which suggested MF intensity-dependent changes in the levels of ROS[31]. We showed that ELF-WMF decreased the mitochondrial ROS level to 81% at 1 h and increased it to 114% at 12 h (Fig. 2c). The inconsistent effects of MF on the levels of ROS in previous reports may be partly accounted for by the temporal profiles of ROS levels.

We demonstrate that the target of Opti-ELF-WMF is the mitochondrial ETC complex II (Fig. 6a, b), but the underlying molecular mechanisms remain to be elucidated. Two models are proposed for the effect of WMF: the ion cyclotron resonance (ICR) effect as the classical mechanism[32,33] and the radical pair model as the quantum mechanism[34,35]. The radical pair model has been applied to the magnetic effect for cryptochrome (Cry). A magnetoreceptor protein (MagR) bound with Cry is also identified, which conducts a nanoscale magnetoreception in many organisms including mammals[36]. The Cry/MagR complex serves as a biocompass in these animals. The mitochondrial ETC complex II and Cry/MagR complex share the same components: flavin adenine dinucleotide (FAD) and iron–sulfur clusters. As mitochondrial ETC complexes I, II, and III have 8, 3, and 1 iron–sulfur clusters, respectively, FAD alone or a combination of FAD and iron–sulfur clusters may account for the effects of ELF-WMF. A moiety in the mitochondrial ETC complex II that is targeted by Opti-ELF-WMF may reside in a structure shared with the Cry/MagR complex.

We show that Opti-ELF-WMF induced mitophagy, followed by upregulation of the mitochondrial ETC activity. Similar to our observation, chemical inhibition of mitochondrial ETC complex II potentially provides neuroprotection by inducing autophagy in cultured neuronal cells[37]. Opti-ELF-WMF may be applicable to human diseases, in which amelioration of compromised mitophagy and enhancement of normal mitophagy would be beneficial.

Repetitive transcranial magnetic stimulation (rTMS) is approved by FDA for treating depression, migraine, and compulsive disorder. Similarly, according to the guidelines of evidence-based medicine for rTMS, level A evidence indicating definite efficacy is reached for depression and stroke-associated motor deficits[38,39]. In addition, level B evidence indicating probable efficacy is reached for Parkinson's disease, multiple sclerosis, fibromyalgia, aphasia, and post-traumatic stress disorder. Although the therapeutc mechanisms of rTMS remain mostly elusive, rTMS preserves mitochondrial membrane integrity in a rat model of ischemic stroke[40] and decreases oxidative stress in a rat model of autoimmune encephalomyelitis[41]. If rTMS and Opti-ELF-WMF share similar mechanisms, Opti-ELF-WMF may serve as a safe alternative to rTMS, in which the magnetic intensities up to 3.0 T potentially induce epilepsy and distressing sensation by stimulating the nerve and muscle[42,43].

We demonstrated that Opti-ELF-WMF induced mitophagy by inhibiting the mitochondrial ETC complex II activity, which was followed by hormetic facilitation of the mitochondrial ETC activity. We evaluated the effects under continuous exposure to Opti-ELF-WMF. However, discontinuation of the exposure or intermittent exposure might have exerted more effects. The

**Table 1 Gene Ontologies (GOs) suppressed by Opti-ELF-WMF in mouse hepatocyte-derived AML12 cells at 1 h by Gene Set Enrichment Analysis (GSEA).**

| GO name | Size | p-value | q-value |
|---|---|---|---|
| GO_NADH_DEHYDROGENASE_ACTIVITY | 43 | $<1 \times 10^{-6}$ | 0.0134 |
| GO_RESPIRATORY_CHAIN_COMPLEX | 73 | $<1 \times 10^{-6}$ | 0.0265 |
| GO_RESPIRASOME | 86 | $<1 \times 10^{-6}$ | 0.0317 |
| GO_MITOCHONDRIAL_ELECTRON_TRANSPORT_NADH_TO_UBIQUINONE | 52 | $<1 \times 10^{-6}$ | 0.0387 |
| GO_OXIDOREDUCTASE_ACTIVITY_ACTING_ON_NAD_P_H_QUINONE_OR_SIMILAR_COMPOUND_AS_ACCEPTOR | 50 | $<1 \times 10^{-6}$ | 0.0465 |
| GO_ATP_SYNTHESIS_COUPLED_ELECTRON_TRANSPORT | 90 | $<1 \times 10^{-6}$ | 0.0492 |
| GO_NADH_DEHYDROGENASE_COMPLEX | 47 | $<1 \times 10^{-6}$ | 0.0713 |
| GO_RESPIRATORY_ELECTRON_TRANSPORT_CHAIN | 106 | $<1 \times 10^{-6}$ | 0.1468 |
| GO_OXIDATIVE_PHOSPHORYLATION | 132 | $<1 \times 10^{-6}$ | 0.1661 |

temporal profile of Opti-ELF-WMF might be able to be optimized in the future.

## Methods

**The ELF-WMF device**. The ELF-WMF device was manufactured by Mr. Kota Okada at the Technical Center of Nagoya University, Japan. The device had a round coil (1 cm height, 10.5 cm inner diameter, and 10.7 cm outer diameter, 50 turns of copper wire of 0.29 mm diameter). The current controller could generate the pulse width from 1 to 16 ms, magnetic flux intensity from 0 to 300 μT, and the pulse frequency from 1 to 16 Hz. We used a 10 μT magnetic field of 4 ms pulse width with increasing frequencies of 1, 2, 3, 4, 5, 6, 7, and 8 Hz in 8 s (Opti-ELF-WMF), unless indicated otherwise. This condition minimizes the hysteresis of the electronic resistance of modified Ringer's solution[20,21]. Before and after each experiment, we confirmed the intensity of the magnetic flux using a pulse magnetic field meter (Aichi Micro Intelligent). To reduce the effects of electromagnetic fields generated by an incubator that had small motors at the top and the bottom of the device, as well as by the geomagnetic field, two 5-mm thick copper plates were placed above and below the culture dish in a humidified incubator with 5% CO₂ at 37 °C (Supplementary Fig. 1a). The culture dish was placed directly on the coil, and the coil was placed 4 cm above the bottom copper plate. The two copper plates reduced the SMF at the culture dish in the incubator from 28 to 14 μT. Control cells were prepared simultaneously with ELF-WMF-stimulated cells from a single batch of cultured cells, and were analyzed in parallel in an identical setup to ELF-WMF-stimulated cells in another incubator of the same model, but with the ELF-WMF stimulator turned off. We confirmed that the application of the ELF-WMF stimulus of 300 μT for 24 h did not change the temperature of the culture medium in a 10 cm culture dish placed above the coil by a digital thermometer at 0.1 °C resolution (SN3000, Netsuken).

**Exposure of wild-type mice to Opti-ELF-WMF**. All the studies on mice were approved by the Animal Care and Use Committee of Nagoya University, and were conducted in accordance with the relevant guidelines. Seven-week-old C57BL6/N male mice were purchased from Japan SLC. Two Opti-ELF-WMF devices were placed in tandem beneath the mouse cage (Supplementary Fig. 1c). The intensities of magnetic fields, to which the mouse body was exposed, were from 5.3 to 14.2 μT above the coil (Supplementary Fig. 1d). Four to five mice were housed in a single cage for 4 weeks, and a total of 14 mice were analyzed for each of the Opti-ELF-WMF and control groups. For the control group, the cage was placed above the Opti-ELF-WMF device, but the switch was turned off. To evaluate the effect of Opti-ELF-WMF in a conventional environment, we did not use 5 mm thick copper plates in the mouse study.

**Test for open-field locomotor activity in mice**. Open-field locomotor activity was evaluated using a photometric actimeter (45 cm × 45 cm, IR Actimeter, Panlab). Fast and slow movements were monitored with a grid of infrared beams every 30 min for 24 h and were used as indices for locomotor activity. Eight mice were individually acclimated to the open-field locomotor activity test for 24 h. The mice were divided into two groups so that the average locomotive activities became similar between the two groups. To examine the effects of Opti-ELF-WMF on the locomotor activity in mice, fast and slow movements were measured before (week 0) and after (week 4) exposure. All data were collected using the SEDACOM software (Panlab). Each mouse was tested individually and had no contact with other mice.

**Isolation of mitochondria from the mouse liver**. For the TMRM assay and the measurement of basal OCR, fresh mitochondria isolated from the mouse liver were examined, as described previously[44]. Briefly, a piece of liver was rinsed, minced, and disrupted in a mitochondrial isolation buffer (70 mM sucrose [Wako], 210 mM mannitol [Sigma], 5 mM HEPES [Dojindo], 1 mM EGTA [Sigma], and 0.5% [w/v] fatty acid-free BSA [Sigma], pH7.2) using a homogenizer. The homogenate was centrifuged at 800 × *g* for 10 min at 4 °C. The supernatant was then centrifuged at

8000 × *g* for 10 min at 4 °C. The pellet was suspended in the mitochondrial isolation buffer to obtain the mitochondrial fraction.

**TMRM assay of the isolated mitochondria**. The mitochondrial membrane potential of isolated mitochondria was analyzed using Tetramethylrhodamine, Methyl Ester, Perchlorate (TMRM) (T668, Thermo Scientific), following the procedure using a flow cytometer as described previously[45]. Briefly, the fresh isolated mitochondria were incubated with 100 nM TMRM in the mitochondrial isolation buffer for 30 min at 37 °C in a humidified incubator. The signal intensities of TMRM were quantified by FACSCalibur Flow Cytometer (BD Biosciences). Data were analyzed with CellQuest Pro (BD Biosciences).

**Measurement of basal OCR of the isolated mitochondria**. The basal OCR of the isolated mitochondria (20 μg of mitochondrial proteins per well) isolated from the mouse liver was determined using the Seahorse XFp Extracellular Flux Analyzer (Agilent Technologies). The assay was conducted as described previously[46].

**Cell culture**. Mouse hepatocyte-derived AML12 cell line was purchased from ATCC and cultured in DMEM/F-12 medium (Gibco) with 10% fetal bovine serum (FBS, Thermo Scientific), dexamethasone (Wako), and insulin-transferrin-sodium selenite (Sigma). HeLa, HEK293, Neuro2a, and C2C12 cells were also purchased from ATCC, and were cultured in DMEM (Gibco) with 10% FBS. PINK1 KO HeLa cells were kindly provided by Dr. Richard J. Youle from the National Institute of Neurological Disorders and Stroke[22], and were cultured in DMEM (Gibco) with 10% FBS. Human iPS cells were kindly provided by Dr. Toshiyuki Araki at the National Center of Neurology and Psychiatry, Japan, and were cultured on a plate coated with iMatrix-511 (892011, Takara Bio) in StemFit (AK02N, Ajinomoto) containing 125 ng/ml puromycin and 10 μM Y-27632 (030-24021, Wako Chemical). All cells were at ~60% confluency at 0 h of the Opti-ELF-WMF exposure, and did not reach 100% confluency in 24 h.

**MitoSOX, MitoTracker Green, and TMRM assays of cultured cells**. AML12 cells exposed to Opti-ELF-WMF for the indicated time periods were washed with PBS. MitoSOX (M36008, Thermo Scientific) and MitoTracker Green (M7514, Thermo Scientific) were dissolved in Hank's balanced salt solution (HBSS, Gibco) at 5 μM and 50 nM, respectively. TMRM was dissolved in the medium at 200 nM. Each dye was added to the cells and incubated for 30 min at 37 °C in a humidified incubator. The cells were then washed with PBS, trypsinized, resuspended in PBS, and harvested. Signal intensities of MitoSOX, MitoTracker Green, and TMRM were quantified by FACSCalibur Flow Cytometer (BD Biosciences) according to the manufacturer's protocols. Data were analyzed with CellQuest Pro (BD Biosciences).

**Western blot analysis of cell lysates**. Cultured cells or minced frozen mouse liver were lysed in PLC buffer containing 50 mM HEPES (pH 7.0), 150 mM NaCl, 10% glycerol, 1% TritonX-100, 1.5 mM MgCl₂, 1 mM EGTA, 100 mM NaF, 10 mM sodium pyrophosphate, 1 μg/μl aprotinin, 1 μg/μl leupeptin, 1 μg/μl pepstatin A, and 1 mM PMSF. The cell lysates were rotated at 4 °C for 20 min and centrifuged at 17,900 × *g* at 4 °C for 15 min. The supernatant was incubated at 37 °C for 1 h to analyze the mitochondrial ETC complex proteins or at 95 °C for 5 min to analyze other proteins in the sample buffer (62.5 mM Tris-HCl pH 6.8, 2% SDS, 10% glycerol, 0.005% bromophenol blue, and 2% 2-mercaptoethanol). For LC3-II, the lysates were separated by Tricine-SDS-PAGE on a 16% polyacrylamide gel[47]. For the other proteins, the lysates were separated by Tris-SDS-PAGE on a 10, 12, or 14% SDS-polyacrylamide gel. The samples were then transferred to a poly-vinylidene fluoride membrane (Immobilon-P, Millipore). Membranes were washed in Tris-buffered saline containing 0.05% Tween 20 (TBS-T) and blocked for 1 h at 24 °C in TBS-T with 5% skimmed milk. The membranes were then incubated overnight at 4 °C with specific antibodies, as indicated below. The membranes were washed with TBS-T and incubated with secondary anti-goat IgG (1:2000, sc-2094,

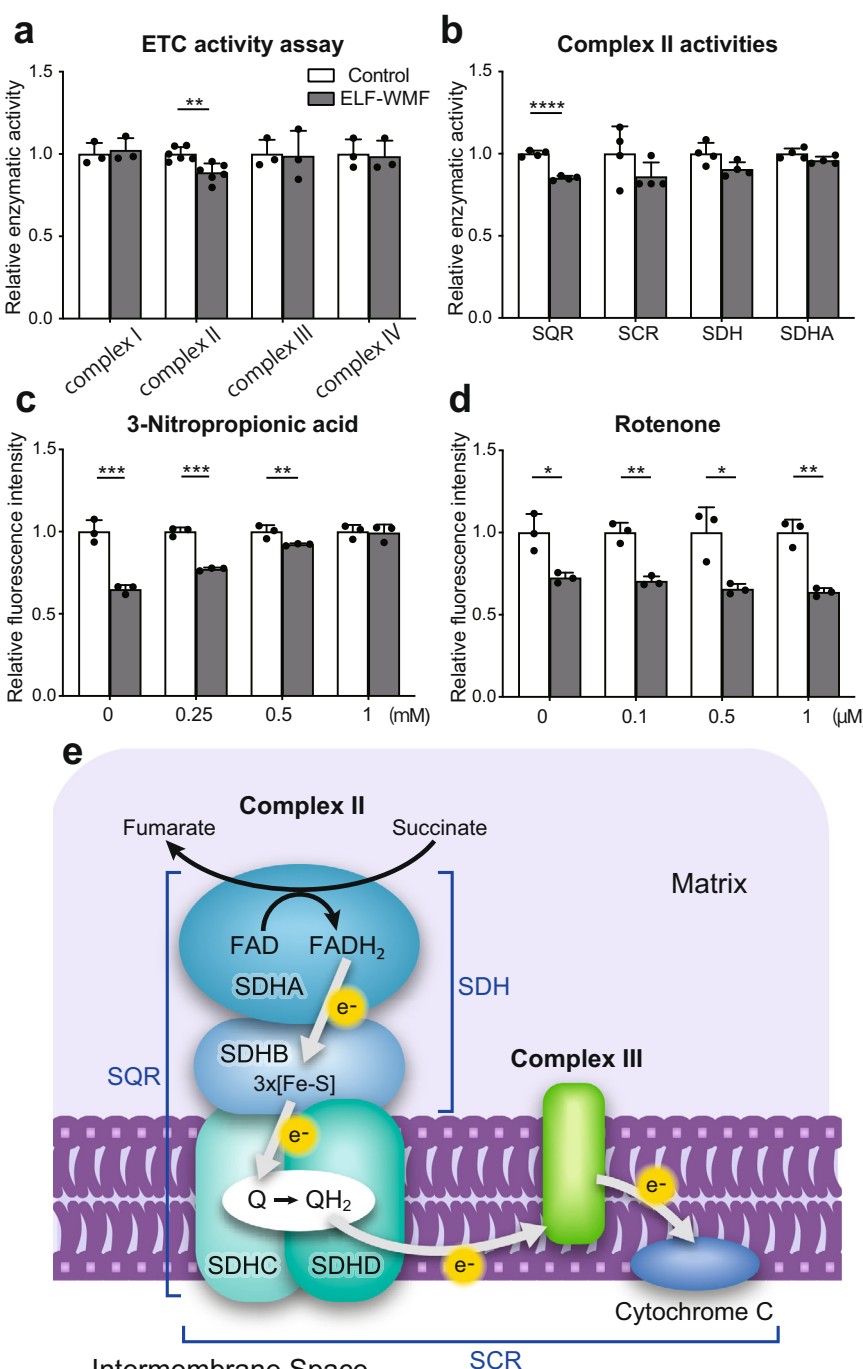

**Fig. 6 Opti-ELF-WMF suppressed the activities of the mitochondrial electron transport chain (ETC) complex II. a** Relative enzymatic activities of mitochondrial ETC complex I, II, III, and IV of mouse liver homogenates exposed to Opti-ELF-WMF in vitro for 8 min (mean ± SD, $n = 3$ to 6 mice each; **$q < 0.01$ by multiple Student's $t$-tests). **b** Fractional and extended ETC complex II activities (succinate:quinone reductase [SQR], succinate-cytochrome $c$ reductase [SCR], succinate dehydrogenase [SDH], succinate dehydrogenase flavoprotein subunit [SDHA]) of isolated mitochondria exposed to Opti-ELF-WMF in vitro for 8 min (mean ± SD, $n = 4$ mice each; ****$q < 0.0001$ by multiple Student's $t$-tests). See **e** for a schematic of the fractional enzymatic activities. **c** Mitochondrial mass (MitoTracker Green) of AML12 cells treated with variable concentrations of a mitochondrial ETC complex II inhibitor, 3-nitropropionic acid, exposed to Opti-ELF-WMF *in cellulo* (mean ± SD, $n = 3$ culture dishes each; **$q < 0.01$ and ***$q < 0.001$ by multiple Student's $t$-tests). **d** Mitochondrial mass (MitoTracker Green) of AML12 cells treated with variable concentrations of a mitochondrial ETC complex I inhibitor, rotenone, exposed to Opti-ELF-WMF *in cellulo* (mean ± SD, $n = 3$ culture dishes each; *$q < 0.05$, and **$q < 0.01$ by multiple Student's $t$-tests). **e** Schematic to indicate the fractional and extended enzymatic activities of mitochondrial ETC complex II measured in **b**. Complex II is composed of SDHA, SDHB, SDHC, and SDHD. See also Fig. S5.

Santa Cruz), anti-mouse IgG (1: 2000, LNA931V/AG, GE Healthcare), or anti-rabbit IgG (1: 2000, LNA934V/AE, GE Healthcare) antibody conjugated to horseradish peroxidase (HRP) for 1 h at 24 °C. Immunoreactive signals were detected with the ECL Western blotting detection reagents (GE Healthcare) and visualized using LAS 4000mini (GE Healthcare). Signal intensities were quantified using ImageQuant (GE Healthcare).

**Antibodies for Western blot analysis**. The following specific antibodies were used for Western blot analysis: anti-UQCRFS1 (1:1000 dilution, ab14746, Abcam), anti-NDUFS1 (1:1000, ab169540, Abcam), anti-VDAC1 (1:3000, ab14734, Abcam), OXPHOS cocktail (1:1000, ab110413, Abcam), anti-LC3 (1:1000, ab51520, Abcam), anti-PINK1 (1:500, ab23707, Abcam), anti-parkin (1:500, #4211, Cell Signaling Technology), anti-ubiquitin (1:1000, P4D1, BioLegend), anti-PGC-1α (1:1000, ab106814, Abcam), anti-PPARα (1:1000, GTX101098, GeneTex), anti-TFAM (1:1000, GTX103231, GeneTex), anti-ATP5A (1:1000, ab14748, Abcam), anti-SDHA (1:1000, GTX101689, GeneTex), anti-SDHB (1:2000, GTX104628, GeneTex), anti-SDHC (1:500, ab155999, Abcam), and anti-SDHD (1:500, ab189945, Abcam) antibodies.

**Preparation of mitochondrial and cytosolic fractions of cultured cells**. A mitochondria isolation kit (ab110170, Abcam) was used for the extraction of mitochondrial and cytosolic fractions according to the manufacturer's protocols. After obtaining the mitochondrial fraction by centrifugation at 12,000 × g for 10 min at 4 °C, the supernatant was used as the cytosolic fraction.

**Detection of mitophagy in cultured cells**. To detect mitophagy in AML12 cells, the Mitophagy Detection Kit (Dojindo Molecular Technologies) was used according to the manufacturer's protocol. Briefly, Mtphagy Dye (Dojindo) dissolved in HBSS at 100 nM was added to the cells and incubated for 30 min at 37 °C in a humidified incubator. After incorporation of Mtphagy Dye into AML12 cells, the cells were exposed to Opti-ELF-WMF for 120, 150, and 180 min, washed with PBS, trypsinized, resuspended in PBS, and harvested. Signal intensities of Mtphagy Dye were quantified by BD FACSCalibur Flow Cytometer (BD Biosciences) according to the manufacturer's protocols. Data were analyzed with CellQuest Pro (BD Biosciences). To visualize both mitophagy and lysosomes after exposure to Opti-ELF-WMF, Lyso Dye (Dojindo) dissolved in HBSS at 1 μM was also added to the cells and incubated for 30 min at 37 °C in a humidified incubator. Images were obtained using a confocal microscope TiE-A1R (Nikon).

**Enzyme assay for mitochondrial ETC complex (I, II, III, and IV) activities of the mouse liver homogenates**. Mitochondrial ETC complex activities were measured using homogenates of the frozen liver excised from C57BL/6N mice. The ETC complex activity assay was performed as previously described[48]. Briefly, the protein concentration of each sample was measured using a Pierce 660 nm protein assay reagent. The ETC complex activities of complexes I, II, III, and IV were estimated by determining the rotenone-sensitive decrease in absorbance of NADH at 340 nm, the decrease in absorbance of 2, 6-dichlorophenolindophenol (DCPIP) at 600 nm, the increase in absorbance of reduced cytochrome c at 550 nm, and the decrease in absorbance of reduced cytochrome c at 550 nm, respectively, with NanoDrop ONE^C (Thermo Scientific). The ETC complex activities were measured in the liver homogenate of mice that were exposed to either Opti-ELF-WMF or control in vivo. Similarly, the ETC complex activities were measured in the mouse liver homogenates before and after exposure to Opti-ELF-WMF for 8 min in vitro.

**Enzyme assay for mitochondrial ETC complex II activity of the mitochondria isolated from the mouse liver**. Mitochondria (10 μg protein) isolated from the liver of C57BL/6 N mice were used to measure the mitochondrial ETC complex II activities (Fig. 6e). Assays for measuring the fractional and extended mitochondrial ETC complex II activities of SQR, SCR, and SDH were performed as previously described[48,49]. The mitochondrial ETC complex II activities were quantified before and after exposure to Opti-ELF-WMF for 8 min in cellulo. The SQR, SCR, and SDH activities were measured by determining the decrease in absorbance of DCPIP at 600 nm, the increase in absorbance of reduced cytochrome c at 550 nm, and the decrease in absorbance of DCPIP at 600 nm, respectively, using NanoDrop ONE^C (Thermo Scientific). The SDHA activity was quantified by modifying a method used to measure the SDH (SDHA and SDHB) activity. For measuring the SDHA activity, 10 μg of sonicated mitochondrial fraction was resuspended in 35 mM phosphate buffer (pH 7.3) supplemented with 0.3 mM KCN (Wako), 10 μg/ml antimycin A (Sigma), 4 mM succinate (Wako), 1.6 mM phenazine methosulfate (PMS) (Sigma), and 40 μM DCPIP (Sigma). The SDHA activity was quantified before and after exposure to Opti-ELF-WMF for 8 min by determining the decrease in absorbance of DCPIP at 600 nm with NanoDrop ONE^c (Thermo Scientific).

**Measurement of mitochondrial mass in AML12 cells exposed to an inhibitor of mitochondrial ETC complex I or II**. AML12 cells were cultured either with variable concentrations of rotenone (Tokyo Chemical Industry Co.), an inhibitor of mitochondrial ETC complex I, or 3-nitropropionic acid (Cayman Chemical), an

inhibitor of mitochondrial ETC complex II, for 12 h. The cells were then exposed to Opti-ELF-WMF for 3 h. Mitochondrial mass was measured by MitoTracker Green, as described above.

**RNA-sequencing and GSEA of AML12 cells**. Total RNA was extracted from AML12 cells exposed to OPTI-ELF-WMF for 1 h using QuickGene-Mini80 (Kurabo) according to the manufacturer's instructions. The extracted RNA was subjected to RNA-seq at Macrogen, Japan. Briefly, a sequencing library was prepared using the TruSeq Stranded mRNA kit (Illumina), and the library was read on an Illumina NovaSeq 6000 (150 bp paired-end reads). GSEA was conducted with the GSEA v4.1.0 software for Windows (https://www.gsea-msigdb.org/gsea/downloads.jsp) using the RNA-seq dataset. RNA-seq data were deposited in the gene expression omnibus (GEO) with an accession number GSE166811.

**Statistics and reproducibility**. All values are presented as the mean ± SD. Statistical significance was estimated either by p-value by Student's t-test, p-value by one-way ANOVA followed by Dunnett's posthoc test, or false discovery rate (q-value) of multiple Student's t-tests. P-values and q-values less than 0.05 were considered statistically significant. The numbers of replicates are indicated in each figure legend.

**Reporting summary**. Further information on research design is available in the Nature Research Reporting Summary linked to this article.

## Data availability
Additional data related to this paper are available upon request to the authors. RNA-seq data were deposited in the Gene Expression Omnibus with an accession number GSE166811. The original, uncropped blot images can be found in Supplementary Fig. 6. Source data can be found in the Supplementary Data.

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

## Acknowledgements

We would like to acknowledge the staff at the Research Core Facility of the Nagoya University Graduate School of Medicine for technical assistance; Mr. Kota Okada at the Technical Center of Nagoya University for fabricating the ELF-WMF devices; Dr. Takamasa Ishii at Tokai University School of Medicine for critical discussion on measuring the activities of subfractions of mitochondrial ETC complex II; and Dr. Richard J. Youle at the National Institute of Neurological Disorders and Stroke for providing us with PINK1KO HeLa cells. This study was supported by Grants-in-Aid from the Japan Society for the Promotion of Science (JP20K06925 to M.I., JP18K14684 to K.O., and JP20H03561 to K.O.); the Ministry of Health, Labor and Welfare of Japan (20FC1036 to K.O.); the Japan Agency for Medical Research and Development (JP21gm1010002 to K.O., JP21ek0109488 to K.O., and JP21bm0804005 to K.O.), the National Center of Neurology and Psychiatry (2–5 to K.O.), the Hori Sciences and Arts Foundation (K.O.), and the Watanabe Foundation (K.O.).

## Author contributions

Conceptualization: K.M. and K.O.; methodology: T.T., M.I., A.M., N.H., and K.O.; investigation: T.T. and M.I.; supervision: H.M., K.M., and K.O.; writing: T.T., M.I., and K.O.

## Competing interests

The authors declare no competing interests.
