## [Peer Review File · Communications Biology]

Reviewers' comments:

Reviewer #1 (Remarks to the Author):

In this manuscript, the authors investigated the influence of the ELF-WMF on mitochondria and found that a specific type of an ELF-WMF reduced mitochondrial mass, membrane potential and ETC activity. Moreover, this ELF-WMF could induce mitophagy. The current version have some major defects that have prevented it from publication, at least not in the current form.

Major points:

1. For the experiment setup, why the authors do not provide a picture of the device? Or at least an illustration. More importantly, it seems from the description that the control condition is not a strict control. The author said that they have used two 5-mm thick copper plates to reduce the effects of electromagnetic field generated by an incubator and the geomagnetic field. The authors should measure the magnetic field before and after using these copper plates. How about the control group? Did they also used two 5-mm thick copper plates? What are the intensity and frequency of the electromagnetic fields generated by the two incubators? Did the two incubators with or without the ELF-WMF device generate identical electromagnetic fields? For such a weak magnetic field investigated in this study, the experimental conditions are very crucial. The environmental electromagnetic fields could easily lead to false results and conclusions.
2. The results in Figure 1b and 2c seem to be conflicting. The authors should discuss about this point.
3. They authors have used different time points in different assays without clear explanations.
4. The relationship between the findings in is study and the diseases stated in the manuscript sounds far-fetched. It has been overstated.

Minor points:

1. The statement of "frequencies of 1–16 Hz every second" seems wrong.
2. In the first paragraph of the introduction, suggesting that "in cellulo" should be "in vitro".
3. The author described that all values are presented as the mean \pm SEM, but the mean \pm SDs were used in many figures.
4. p value and q value were simultaneously used in many figures. Please explain why the q values are necessary.
5. In Figure S2, what is the "red bars"?
6. "western blot" should be "Western blot"

Reviewer #2 (Remarks to the Author):

The paper by Toda et al., addresses a very intriguing hypothesis of a weak magnetic field-triggered mitophagy and rejuvenation to maintain mitochondria homeostasis. The topic is novel and focuses on the biological response of cells –mainly in terms of mitochondria metabolism- to weak magnetic field, a research topic that is mostly uncovered.

Despite these premises, the major criticism to be addressed to the manuscript relies in the exposure setup and exposure control of both in vitro and in vivo experiments. This is a key issue as it weakens the huge –but valuable- amount of experimental data provided in the manuscript.

Exposure system adopted in the in vivo experiments needs to be better characterized to directly correlate the specific biological response to the applied magnetic field.

- Which is the direction of the currents in the two loops placed under the exposure cage?
- Did the authors measure the B-field values in the exposure volume to provide an average B-field value with a standard deviation as a measure of the homogeneity of the B-field in the exposure volume?

All these information are mandatory, without these data, it is impossible to accept the paper.

Even the exposure system adopted in the in vitro experiments needs to be better characterized:

- which kind of flask or Petri-dish did the authors use to perform experiments with cells?
- which volume of culture media was used?
- which was the position of the cells within flasks/dishes with respect with the geometry of the coils in the incubator? Maybe a scheme of the exposure setup might be useful.
- Did the authors measure the B-field values in the exposure volume to provide an average B-field value with a standard deviation as a measure of the homogeneity of the B-field in the exposure volume?

Even in this case these details are mandatory, especially when so many different exposure conditions were tested and so many different molecular and biochemical pathways were investigated.

There is no mention of the methods and devices used to control temperature over the exposure period.

There is no use of a proper Sham control: the ideal exposure experimental plan should include control sample (not exposed, either switched-off system or different cage and incubator), sham-exposed sample (exposure to the system in the "no magnetic field" condition that might be achieved, for instance, by allowing the currents to flow within the coils in opposite directions), and MF-exposed sample.

Did the authors perform the exposures under blind condition?

Other comments:

- Mice experiments. How was the statistical power calculated and the number of N=4 decided? Were the mice randomly assigned? Were the mice and liver weights calculated at the end of the exposure?
- Mouse liver mitochondria isolation. How were the mitochondria stored until analyses? Were the TMRM, the basal oxygen consumption and the ETC complex analyses performed on fresh or frozen mice mitochondria samples???? It is not clear the way the samples were split to allow all kind of tests.
- Some more details about the flow cytometer analyses should be provided, such as the methodology the TMRM, mitosox, mitotracker were quantified.
- In western blot analyses of mitochondria proteins, a mitochondria-specific protein might be more informative for normalization in addition to beta-actin, also to provide evidence of the purity of the mitochondria isolation.
- Cell cultures. There is no information about culture condition in incubators, about the passage number the cells were used in the experiments. Not all readers are familiar with the names of cell cultures, therefore the specification of cell type might be useful, such as "AML12 mice hepatocytes" instead of simply "AML12 cells".
- Experiments reported in table 1 seem to suggest that the effect on mitochondria is not cell-specific, but the table is incomplete in the present form as it reports mitochondria mass at 3h and membrane potential at 12h, whereas both endpoints should be measured at both 3 and 12 h to sustain what the authors claim in the results (lines 319-322).

Minor comments:

- A list of the abbreviations used throughout the manuscript might be useful

Reviewer #4 (Remarks to the Author):

This paper focuses on the effect of extremely low-frequency pulses of faint magnetic field, weaker than the geomagnetic field on mitochondria by analyzing several biological endpoints. The results are quite interesting, but the manuscript needs a revision to make it suitable for publication. Some specific suggestions are reported below.

- 1) Static weak magnetic fields needed to be emphasized when it comes to report increase in calcium levels at intracellular reports. More references can be added. See below.

Bekhite, M.M., Figulla, H.R., Sauer, H. and Wartenberg, M., 2013. Static magnetic fields increase cardiomyocyte differentiation of Flk-1+ cells derived from mouse embryonic stem cells via Ca²⁺ influx and ROS production. *International journal of cardiology*, 167(3), pp.798-808.

2) There are papers in vitro you could find reports of static magnetic fields weaker than geomagnetic field. See below.

Novikov, V. V., E. V. Yablokova, I. A. Shaev, and E. E. Fesenko. "The Effect of a Weak Static Magnetic Field in the Range of Magnitudes from a "Zero" Field (0.01 μ T) to 100 μ T on the Production of Reactive Oxygen Species in Nonactivated Neutrophils." *Biophysics* 65, no. 3 (2020): 443-447.

Gurhan, H., Bruzon, R., Kandala, S., Greenebaum, B. and Barnes, F., 2021. Effects induced by a weak static magnetic field of different intensities on HT-1080 fibrosarcoma cells. *Bioelectromagnetics*, 42(3), pp.212-223.

3) Radical Pair Mechanism and Cyclotron Resonance are couple examples of molecular mechanisms, and it is worth to mention.

4) 5-mm thick copper plates are not sufficient to cancel out earth magnetic field. I would recommend using a mu metal box with holes to allow air flow.

5) A figure showing the exposure setup could be useful.

6) Temperature measurements with and without the magnetic field exposure needs to be made. And results need to be discussed whether temperature variations make an impact on cell growth or chemical activities of interest in cell.

7) What passage numbers of cells are being used? Experiments with control and treated units were run with using same passages, identical incubator? Even if you are running experiments in another incubator of same brand you need to calibrate temperature and CO₂ levels frequently.

8) A discussion about given exposure times could be useful. When do cells reach confluence and what are the reasoning behind chosen exposure times?

We cordially appreciate scrutinizing comments by the reviewers. Our specific responses are listed below and the revisions are highlighted in the revised manuscript.

Comments by Reviewer #1

Comment Rev1_1

For the experiment setup, why the authors do not provide a picture of the device? Or at least an illustration. More importantly, it seems from the description that the control condition is not a strict control. The author said that they have used two 5-mm thick copper plates to reduce the effects of electromagnetic field generated by an incubator and the geomagnetic field. The authors should measure the magnetic field before and after using these copper plates. How about the control group? Did they also used two 5-mm thick copper plates? What are the intensity and frequency of the electromagnetic fields generated by the two incubators? Did the two incubators with or without the ELF-WMF device generate identical electromagnetic fields? For such a weak magnetic field investigated in this study, the experimental conditions are very crucial. The environmental electromagnetic fields could easily lead to false results and conclusions.

Response to Rev1_1

We appreciate the suggestion. We showed experimental setup of the coil and the controller in Fig. S1abc. We showed the effect of copper plates on the environmental static magnetic field. Control cells and ELF-WMF-exposed cells were prepared from the same batch of cultured cells. Control cells were placed in another incubator with the same setup but with turning off the ELF-EWF controller. We added the following statements in Materials and Methods.

Added statements in Materials and Methods

The ELF-WMF apparatus

To reduce the effects of electromagnetic fields generated by an incubator that had small motors at the top and the bottom of the device, as well as by the geomagnetic field, two 5-mm thick copper plates were placed above and below the culture dish in a humidified incubator with 5% CO₂ at 37°C (Fig. S1a). The culture dish was placed directly on the coil, and the coil was placed 4 cm above the bottom copper plate. The two copper plates reduced the static magnetic field at the culture dish in the incubator from 28 μT to 14 μT. When the stimulation intensity was set to 10 μT, the pulsative magnetic field intensity at the base of the culture dish was 9.95 ± 1.02 μT (mean and SD, *n* = 100 measurements). Control cells were prepared simultaneously with ELF-WMF-stimulated cells from a single batch of cultured cells, and were analyzed in parallel in an identical setup to ELF-WMF-stimulated cells in another incubator of the same model, but with the ELF-WMF stimulator turned off. We confirmed that application of the magnetic field intensity from 0 to 300 μT for 24 h had no effect on the temperature of the culture medium at 0.1°C resolution.

Revised statements in Figure S1

Figure S1. The setup of the ELF-WMF device and lack of the effect of Opti-ELF-WMF for 4 weeks on the open-field locomotor activities in wild-type mice, related to Figure 1. **a** ELF-WMF setup for cultured cell. Four of the six wells were used for cell culture. All cells were fit within a 10.5-cm coil. Cells were sandwiched by copper plates, and the coil was placed 4 cm above the bottom plate. **b** Electric current controller for the ELF-WMF coil. **c** Two ELF-WMF coils were placed directly beneath a cage for mouse studies. Exposure to Opti-ELF-WMF for 4 weeks had no effect on fast **(d)** and slow **(e)**

movements of wild-type mice as determined by an open-field locomotor activity test (mean \pm SD, $n = 4$ mice each; no statistical [n.s.] difference by Kruskal–Wallis test).

Comment Rev1_2

The results in Figure 1b and 2c seem to be conflicting. The authors should discuss about this point.

Response to Rev1_2

We apologize for confusing statements. Fig. 1b showed the effect of ELF-WMF for 4 weeks on the mouse liver. In contrast, Fig. 2c showed the effect of ELF-WMF for 0-24 h on AML12 cells. We did not examine whether the mitochondrial membrane potential was decreased or not in 6 h in the mouse liver, as we observed in AML12 cells. We clarified that we did not examine the acute effect of ELF-WMF on the mouse liver mitochondria in Discussion.

Revised statements in Discussion

We also showed that Opti-ELF-WMF for 4 weeks increased the mitochondrial membrane potential in the mouse liver by ~40% (Fig. 1b), although we did not examine the acute effect of Opti-ELF-WMF on the mouse liver mitochondria.

Comment Rev1_3

They authors have used different time points in different assays without clear explanations.

Response to Rev1_3

We repeatedly performed each assay in a range of different time frames in AML12 cells, and confirmed that ELF-WMF first suppressed ETC II activity, and then induced mitophagy and mitochondrial genesis. We only showed the results in the optimal time frame for each assay.

For ETC complex activities, we consistently used 8-min incubation time throughout our manuscript. We revised Fig. 6a and the relevant statement in Materials and Methods.

Revised statement in Materials and Methods

Enzyme assay for mitochondrial ETC complex (I, II, III, IV) activities of the mouse liver homogenates

Similarly, the ETC complex activities were measured in the mouse liver homogenates before and after exposure to ELF-WMF for 8 min *in vitro*.

Comment Rev1_4

The relationship between the findings in is study and the diseases stated in the manuscript sounds far-fetched. It has been overstated.

Response to Rev1_4

We appreciate your critical suggestion. We have preliminary data showing the effects of ELF-WMF on models of Parkinson's disease and mitochondrial diseases, but we did not show any relevant data. We agreed that we overstated the effects of ELF-WMF on these diseases. As suggested, we eliminated statements on specific diseases in Discussion.

Minor Comment Rev1_5

The statement of “frequencies of 1–16 Hz every second” seems wrong.

Response to Rev1_5

Thank you for pointing this out. We clarified our statement in Materials and Methods.

Revised statement in Materials and Methods

The ELF-WMF apparatus

The current controller could generate the pulse width from 1 to 16 ms, magnetic flux intensity from 0 to 300 μ T, and the pulse frequency from 1 to 16 Hz.

Minor Comment Rev1_6

In the first paragraph of the introduction, suggesting that “in cellulo” should be “in vitro”.

Response to Rev1_6

As suggested, we changed “*in cellulo*” to “*in vitro*” in Introduction.

Minor Comment Rev1_7

The author described that all values are presented as the mean \pm SEM, but the mean \pm SDs were used in many figures.

Response to Rev1_7

We apologize for our inadvertent mistake. We corrected our statement in Materials and Methods.

Revised statement in Materials and Methods

Statistical analysis

All values are presented as the mean \pm SD.

Minor Comment Rev1_8

p value and q value were simultaneously used in many figures. Please explain why the q values are necessary.

Response to Rev1_8

We clarified that p -value is for Student’s t -test and one-way ANOVA with Dunnett’s posthoc test, and that q -value represents a false discovery rate in Materials and Methods.

Revised statements in Materials and Methods

Statistical analysis

Statistical significance was estimated either by p -value by Student’s t -test, p -value by one-way ANOVA followed by Dunnett’s posthoc test, or false discovery rate (q -value) of multiple Student’s t -tests. P -values and q -values less than 0.05 were considered statistically significant.

Minor Comment Rev1_9

In Figure S2, what is the “red bars”?

Response to Rev1_9

We apologize that we changed color images to a grayscale images. We corrected it to “black bar” in the Figure legend.

Revised legend for Fig. S2

Figure S2. Optimization of ELF-WMF. Optimization of magnetic flux intensity (a), pulse width (b), and pulse frequency (c) of ELF-WMF to reduce the mitochondrial mass (MitoTracker Green) of AML12 cells in 3 h (mean \pm SD, $n = 4$ culture dishes each; * $p < 0.05$, ** $p < 0.01$, *** $p < 0.001$, and **** $p < 0.0001$ by one-way ANOVA followed by Dunnett’s posthoc test between the black bar and the gray bars). Conditions of Opti-ELF-WMF are indicted by black bars.

Minor Comment Rev1_10

“western blot” should be “Western blot”

Response to Rev1_10

Thank you. We corrected them.

Comments by Reviewer #2

Comment Rev2_1

Exposure system adopted in the in vivo experiments needs to be better characterized to directly correlate the specific biological response to the applied magnetic field.

- Which is the direction of the currents in the two loops placed under the exposure cage?
- Did the authors measure the B-field values in the exposure volume to provide an average B-field value with a standard deviation as a measure of the homogeneity of the B-field in the exposure volume?

Response to Rev2_1

Thank you for productive and important suggestions. As suggested, we showed the pictures of our device in Fig. S1. For mouse studies, two coils were placed in parallel beneath the cage (Fig. S1C). The B field value at the base of a culture dish was $9.95 \pm 1.02 \mu\text{T}$ (mean \pm SD, $n = 100$ measurements). We rewrote the device setup in more detail in Materials and Methods.

Revised statements in Materials and Methods

The ELF-WMF apparatus

To reduce the effects of electromagnetic fields generated by an incubator that had small motors at the top and the bottom of the device, as well as by the geomagnetic field, two 5-mm thick copper plates were placed above and below the culture dish in a humidified incubator with 5% CO₂ at 37°C (Fig. S1a). The culture dish was placed directly on the coil, and the coil was placed 4 cm above the bottom copper plate. The two copper plates reduced the static magnetic field at the culture dish in the incubator from 28 μT to 14 μT . When the stimulation intensity was set to 10 μT , the pulsative magnetic field intensity at the base of the culture dish was $9.95 \pm 1.02 \mu\text{T}$ (mean and SD, $n = 100$ measurements). Control cells were prepared simultaneously with ELF-WMF-stimulated cells from a single batch of cultured cells, and were analyzed in parallel in an identical setup to ELF-WMF-stimulated cells in another incubator of the same model, but with the ELF-WMF stimulator turned off. We confirmed that application of the magnetic field intensity from 0 to 300 μT for 24 h had no effect on the temperature of the culture medium at 0.1°C resolution.

Revised statements in Figure S1

Figure S1. The setup of the ELF-WMF device and lack of the effect of Opti-ELF-WMF for 4 weeks on the open-field locomotor activities in wild-type mice, related to Figure 1. **a** ELF-WMF setup for cultured cell. Four of the six wells were used for cell culture. All cells were fit within a 10.5-cm coil. Cells were sandwiched by copper plates, and the coil was placed 4 cm above the bottom plate. **b** Electric current controller for the ELF-WMF coil. **c** Two ELF-WMF coils were placed directly beneath a cage for mouse studies. Exposure to Opti-ELF-WMF for 4 weeks had no effect on fast (**d**) and slow (**e**) movements of wild-type mice as determined by an open-field locomotor activity test (mean \pm SD, $n = 4$ mice each; no statistical [n.s.] difference by Kruskal–Wallis test).

Comment Rev2_2

Even the exposure system adopted in the in vitro experiments needs to be better

characterized:

- which kind of flask or Petri-dish did the authors use to perform experiments with cells?
- which volume of culture media was used?
- which was the position of the cells within flasks/dishes with respect with the geometry of the coils in the incubator? Maybe a scheme of the exposure setup might be useful.
- Did the authors measure the B-field values in the exposure volume to provide an average B-field value with a standard deviation as a measure of the homogeneity of the B-field in the exposure volume?

There is no mention of the methods and devices used to control temperature over the exposure period.

Response to Rev2_2

We appreciate the suggestions. For experiments without mitochondrial isolation, we cultured AML12 cells in a 6-well plate with 2 ml medium each. Four out of six wells were used for cell culture, and the center of the four wells was placed above the center of the coil so that all cultured cells were fit within the 10-cm loop (Fig. S1a). The B-field values should be slightly different from cell to cell, but there should be no variability in the four wells. For experiment with mitochondrial isolation, we cultured AML12 cells in a 10-cm Petri dish with 10 ml medium. All cultured cells in a 10-cm dish were fit within the 10-cm loop. For both setups, the mean and SD of B-field values were 9.96 and 1.02 μ T. We confirmed that even 300 μ T ELF-WMF stimulation for 24 h did not change the temperature of the culture medium at 0.1°C resolution. See revised Materials and Methods cited in Rev2_1 above.

Comment Rev2_3

There is no use of a proper Sham control: the ideal exposure experimental plan should include control sample (not exposed, either switched-off system or different cage and incubator), sham-exposed sample (exposure to the system in the “no magnetic field” condition that might be achieved, for instance, by allowing the currents to flow within the coils in opposite directions), and MF-exposed sample.

Response to Rev2_3

We used the same setup for control cells and control mice but with turning off the controller. We used these controls in our initial submission. We clarified how we setup controls in more detail in Materials and Methods.

Revised statements in Materials and Methods

The ELF-WMF device

Control cells were prepared simultaneously with ELF-WMF-exposed cells from a single batch of cultured cells, and were analyzed in parallel in an identical setup to ELF-WMF-exposed cells in another incubator of the same model, but with the ELF-WMF stimulator turned off.

Exposure of wild-type mice to ELF-WMF

All the studies on mice were approved by the Animal Care and Use Committee of Nagoya University, and were conducted in accordance with the relevant guidelines. Seven-week-old C57BL6/N male mice were purchased from Japan SLC. Two ELF-WMF devices were placed in tandem beneath the mouse cage (Fig. S1c). For the ELF-WMF group, four mice were housed in a single cage with switch on for 4 weeks. For the control group, four mice were housed in a single cage with switch off for 4 weeks. The Opti-ELF-WMF condition stated above was applied to the ELF-WMF group.

Comment Rev2_4

Did the authors perform the exposures under blind condition?

Response to Rev2_4

Thank you for commenting on a critical point. Neither cell studies nor mouse studies were blinded. We usually perform blinded experiments, when we have to have subjective measures like visual evaluation of the motor functions and manual scoring of immunostained images. We did not use any subjective measures in this communication. For open-field locomotor activity test, mouse movements were objectively measured by a photometric actimeter, which automatically counts the number of crossing infrared beams.

Comment Rev2_5

- Mice experiments. How was the statistical power calculated and the number of N=4 decided? Were the mice randomly assigned?

Were the mice and liver weights calculated at the end of the exposure?

Response to Rev2_5

We agree that if we had used a large number of mice, we might have detected differences in locomotive activities. However, the Animal Care and Use Committee allowed us to use the minimum number of mice. What we presented in our manuscript was an exploratory study, and not a confirmatory study that is required for formal clinical trial. Therefore, we did not calculate the number of required mice from the effect size of our study. For the locomotive analysis of mice, locomotive activities of eight mice were measured in a day before the examination, and mice were divided into two groups so that the mean locomotive activities became similar. We revised our statements in Materials and Methods. We did not measure the mouse or liver weight.

Revised statements in Materials and Methods**Test for open-field locomotor activity in mice**

Open-field locomotor activity was evaluated using a photometric actimeter (45 cm × 45 cm, IR Actimeter, Panlab). Fast and slow movements were monitored with a grid of infrared beams every 30 min for 24 h and were used as indices for locomotor activity. Eight mice were individually acclimated to the open-field locomotor activity test for 24 h. The mice were divided into two groups so that the average locomotive activities became similar between the two groups. To examine the effects of ELF-WMF on the locomotor activity in mice, fast and slow movements were automatically measured before (week 0) and after (week 4) exposure. All data were collected using the SEDACOM software (Panlab). Each mouse was tested individually and had no contact with other mice.

Comment Rev2_6

- Mouse liver mitochondria isolation. How were the mitochondria stored until analyses? Were the TMRM, the basal oxygen consumption and the ETC complex analyses performed on fresh or frozen mice mitochondria samples???? It is not clear the way the samples were split to allow all kind of tests.

Response to Rev2_6

For the TMRM assay and the measurement of basal oxygen consumption rate by a flux analyzer, fresh mitochondria isolated from the mouse liver were examined. For the

measurements of ETC complex enzyme activities and Western Blotting, the mouse liver was frozen at -80°C before analysis. We indicated these in Materials and Methods.

Revised statements in Materials and Methods

Isolation of mitochondria from the mouse liver

For the TMRM assay and the measurement of basal oxygen consumption rate, fresh mitochondria isolated from the mouse liver were examined, as described previously¹³.

Western blot analysis of cell lysates

Cells or minced frozen mouse liver were lysed in PLC buffer containing 50 mM HEPES (pH 7.0), 150 mM NaCl, 10% glycerol, 1% TritonX-100, 1.5 mM MgCl₂, 1 mM EGTA, 100 mM NaF, 10 mM sodium pyrophosphate, 1 µg/µl aprotinin, 1 µg/µl leupeptin, 1 µg/µl pepstatin A, and 1 mM PMSF.

Enzyme assay for mitochondrial ETC complex (I, II, III, IV) activities of the mouse liver homogenates

Mitochondrial ETC complex activities were measured using homogenates of the frozen liver excised from C57BL/6N mice.

Comment Rev2_7

- Some more details about the flow cytometer analyses should be provided, such as the methodology the TMRM, mitosox, mitotracker were quantified.

Response to Rev2_7

Detailed protocols using a flow cytometer for the TMRM, MitoSox, and MitoTracker Green assays were provided by the manufacturers, and we followed them. We indicated that we followed the manufacturer's protocols. We also indicated that we used CellQuest Pro (BD Biosciences) for the analysis. The revisions were made in multiple sections, and are not pasted here.

Comment Rev2_8

- In western blot analyses of mitochondria proteins, a mitochondria-specific protein might be more informative for normalization in addition to beta-actin, also to provide evidence of the purity of the mitochondria isolation.

Response to Rev2_8

We appreciate the suggestion. We normalized mitochondrial proteins by mitochondria-specific VDAC1, and cellular and cytoplasmic proteins by beta-actin. To indicate the purity of mitochondrial isolation, we added Western blotting of beta-actin and VDAC1 in whole, cytosolic, and mitochondrial fractions in Fig 4b.

Added statement in Results

Next, we isolated the mitochondrial and cytosolic fractions of AML12 cells, and examined the purity by immunoblotting of β-actin and VDAC1, respectively (Fig. 4b).

Added legend for Figure 4b

Figure 4. ELF-WMF induced mitophagy. b Western blot analysis of β-actin and VDAC1 in whole, cytosolic, and mitochondrial fractions of AML12 cells.

Comment Rev2_9

Cell cultures. There is no information about culture condition in incubators, about the passage number the cells were used in the experiments. Not all readers are familiar with the names of cell cultures, therefore the specification of cell type might be useful, such

as “AML12 mice hepatocytes” instead of simply “AML12 cells”.

Response to Rev2_9

We appreciate the suggestion. We always used the cells with the same number of passages for the control and ELF-WMF groups by preparing the two groups from a single batch of cultured cells. See pasted revised statements in Materials and Methods in Response to Rev2_3. As suggested, we indicated “Mouse hepatocyte AML12 cell line” in Materials and Methods, Results, and Figure legends. Each revision is not pasted here.

Comment Rev2_10

- Experiments reported in table 1 seem to suggest that the effect on mitochondria is not cell-specific, but the table is incomplete in the present form as it reports mitochondria mass at 3h and membrane potential at 12h, whereas both endpoints should be measured at both 3 and 12 h to sustain what the authors claim in the results (lines 319-322).

Response to Rev2_10

We appreciate critical and productive suggestion. We used different measures at 3 and 12 h, because in AML12 cells the mitochondrial mass was more reduced than the mitochondrial membrane potential at 3 h, and the mitochondrial membrane potential was more increased at 12 h. We hoped to examine whether what we observed in AML12 cells could be applied to five other cell lines. We agree that some cells may show different temporal profiles in the mitochondrial mass and the mitochondrial membrane potential. However, demonstration of the effects of ELF-WMF on the five additional cells are additive and are not essential in this communication. We thus moved Table 1 to Supplementary Table S2.

Minor Comment Rev2_11

- A list of the abbreviations used throughout the manuscript might be useful

Response to Rev2_11

Thank you for the suggestions. We made a list of abbreviations in Table S1

Table S1. List of abbreviations

Abbreviation	Definition
WMF	weak magnetic fields
ELF	extremely low-frequency
ETC	electron transport chain
ROS	reactive oxygen species
PINK1	PTEN-induced kinase 1
TMRM	Tetramethylrhodamine, Methyl Ester, Perchlorate
OCR	oxygen consumption rate
DCPIP	2, 6-dichlorophenolindophenol
SDH	succinate dehydrogenase
SQR	succinate:quinone reductase
SCR	succinate cytochrome c reductase
SDHA	succinate dehydrogenase [ubiquinone] flavoprotein subunit

SDHB	succinate dehydrogenase [ubiquinone] iron-sulfur subunit
SDHC	succinate dehydrogenase cytochrome b560 subunit
SDHD	succinate dehydrogenase [ubiquinone] cytochrome b small subunit

Comments by Reviewer #4

Comment Rev4_1

- 1) Static weak magnetic fields needed to be emphasized when it comes to report increase in calcium levels at intracellular reports. More references can be added. Bekhite et.al , 2013.
- 2) There are papers in vitro you could find reports of static magnetic fields weaker than geomagnetic field. Novikov et.al , 2020, Gurhan et.al 2021

Response to Rev4_1

Thank you for your kind suggestions. We cited Bekhite et.al , 2013; Novikov et.al , 2020; and Gurhan et.al 2021 in Introduction.

Added statements in Introduction

Introduction

Similarly, static magnetic fields increase cytosolic calcium and reactive oxygen species (ROS) in mouse embryonic stem (ES) cell-derived embryoid bodies and Flk-1+ cardiac progenitor cells², although the magnetic intensities were 0.3 to 5.0 mT. In addition, static WMF as weak as 0.01 μ T reduces the ROS level in nonactivated neutrophils³. Moreover, the exposure to static WMF of 200 to 600 μ T in HT1080 cells increased the mitochondrial calcium concentration and the mitochondrial membrane potential⁴.

Comment Rev4_2

Radical Pair Mechanism and Cyclotron Resonance are couple examples of molecular mechanisms, and it is worth to mention.

Response to Rev4_2

We appreciate the valuable suggestion. As suggested, we addressed two molecular mechanisms of the radical pair model and the cyclotron resonance effect in Discussion.

Added statement in Discussion

Two models are proposed for the effect of WMF: the ion cyclotron resonance (ICR) effect as the classical mechanism^{32, 33} and the radical pair model as the quantum mechanism^{34, 35}. The radical pair model has been applied to the magnetic effect for cryptochrome (Cry).

Comment Rev4_3

5-mm thick copper plates are not sufficient to cancel out earth magnetic field. I would recommend using a mu metal box with holes to allow air flow.

Response to Rev4_3

We appreciate precious suggestions. We observed the effect of ELF-WMF even when the geomagnetic field was not blocked at all. We thus thought that blocking of geomagnetic field was not required for ELF-WMF to exert its effects on cultured cells. We observed that the 5-mm thick copper plates reduced the static and pulsative magnetic fields in an incubator to ~50% and ~20%, respectively. We hoped that the reduction of the effects of pulsative incubator-generated magnetic fields by copper plates would yield reproducible data that were not dependent on a specific incubator. We indicated the effects of the 5-mm copper plates in an incubator in more detail. We appreciate the suggestion of a mu

metal, but we could not afford the expensive mu metal.

Added statements in Materials and Methods

The ELF-WMF device

To reduce the effects of electromagnetic fields generated by an incubator that had small motors at the top and the bottom of the device, as well as by the geomagnetic field, two 5-mm thick copper plates were placed above and below the culture dish in a humidified incubator with 5% CO₂ at 37°C (Fig. S1a). The culture dish was placed directly on the coil, and the coil was placed 4 cm above the bottom copper plate. The two copper plates reduced the static magnetic field at the culture dish in the incubator from 28 μT to 14 μT. When the stimulation intensity was set to 10 μT, the pulsative magnetic field intensity at the base of the culture dish was 9.95 ± 1.02 μT (mean and SD, *n* = 100 measurements). Control cells were prepared simultaneously with ELF-WMF-exposed cells from a single batch of cultured cells, and were analyzed in parallel in an identical setup to ELF-WMF-exposed cells in another incubator of the same model, but with the ELF-WMF stimulator turned off. We confirmed that application of the magnetic field intensity from 0 to 300 μT for 24 h had no effect on the temperature of the culture medium at 0.1°C resolution.

Comment Rev4_4

A figure showing the exposure setup could be useful.

Response to Rev4_4

We appreciate the valuable suggestion. We showed the ELF-WMF setup for cultured cells and mice in Fig. S1abc.

Added legends for Fig. S1abc

Figure S1. The setup of the ELF-WMF device and lack of the effect of Opti-ELF-WMF for 4 weeks on the open-field locomotor activities in wild-type mice, related to Figure 1. **a** ELF-WMF setup for cultured cell. Four of the six wells were used for cell culture. All cells were fit within a 10.5-cm coil. Cells were sandwiched by copper plates, and the coil was placed 4 cm above the bottom plate. **b** Electric current controller for the ELF-WMF coil. **c** Two ELF-WMF coils were placed directly beneath a cage for mouse studies. Exposure to Opti-ELF-WMF for 4 weeks had no effect on fast (**d**) and slow (**e**) movements of wild-type mice as determined by an open-field locomotor activity test (mean ± SD, *n* = 4 mice each; no statistical [n.s.] difference by Kruskal–Wallis test).

Comment Rev4_5

Temperature measurements with and without the magnetic field exposure needs to be made. And results need to be discussed whether temperature variations make an impact on cell growth or chemical activities of interest in cell.

What passage numbers of cells are being used? Experiments with control and treated units were run with using same passages, identical incubator? Even if you are running experiments in another incubator of same brand you need to calibrate temperature and CO₂ levels frequently.

Response to Rev4_5

Thank you for the suggestions. We confirmed that temperature of the culture medium in the coil remained unchanged even at 300-μT intensity for 24h. Control and ELF-WMF cells were simultaneously split from the same batch of cultured cells. Control and ELF-WMF cells were cultured in two incubators of the same brand. We also confirmed that

the temperature and the CO₂ concentration were the same in these incubators. Added statements in Materials and Methods were pasted in Response to Rev4_3 above.

Comment Rev4_6

A discussion about given exposure times could be useful. When do cells reach confluence and what are the reasoning behind chosen exposure times?

Response to Rev4_6

We appreciate the critical comments. We did not allow cells to reach confluency, because characteristics of cultured cells can be changed at the confluency. We indicated that cultured cells did not reach confluency during the observation. We also indicated the reasons how we chose the exposure times in Results.

Added statement in Materials and Methods

Cell culture

All cells were ~60% confluency at 0 h of the ELF-WMF exposure, and did not reach 100% confluency in 24 h.

Revised statements in Results

Opti-ELF-WMF temporarily decreases the mitochondrial ROS levels, mitochondrial mass, and mitochondrial membrane potential in cultured cells

Thus, Opti-ELF-WMF suppressed the mitochondrial ETC activity at 1 h, which was likely followed by elimination and/or inactivation of a subset of mitochondria at 3 to 6 h. The mass and function of mitochondria were then increased at 12 h and returned to normal levels at 24 h. These time points were used for subsequent analyses.

Optimal conditions of ELF-WMF for the reduction of the mitochondrial mass in cultured cells

Next, we analyzed the optimal conditions of ELF-WMF that would reduce the mitochondrial mass at 3 h, based on the results shown in Fig. 2b, by changing the intensity, pulse width, and frequency of ELF-WMF.

** Please ensure you delete the link to your author homepage in this email if you wish to forward it to your coauthors **

Dear Dr. Ohno,

Your manuscript entitled "Extremely low-frequency pulses of faint magnetic field, weaker than the geomagnetic field, induce mitophagy to rejuvenate mitochondria" has now been seen by 2 referees. You will see from their comments below that while they find your work of considerable interest, some important points are raised. We are interested in the possibility of publishing your study in *Communications Biology*, but would like to consider your response to these concerns in the form of a revised manuscript before we make a final decision on publication.

We therefore invite you to revise and resubmit your manuscript, taking into account all of the points raised by the reviewers. In particular, we ask that you address all comments regarding experimental design, ethics, number of replicates and controls used (Reviewers 1 and 3). Reviewer 2 was unable to submit their revised report, but we ask that you address the following remaining points if possible:

- state direction of currents in two loops of exposure cage
- add mitochondria specific VDAC1 to Figure 4c
- measure mitochondrial mass and mitochondrial membrane potential at 3h and 12h

Please highlight all changes in the manuscript text file.

We are committed to providing a fair and constructive peer-review process. Do not hesitate to contact us if you wish to discuss the revision in more detail or if there are specific requests from the reviewers that you believe are technically impossible or unlikely to yield a meaningful outcome.

At the same time, we ask that you ensure your manuscript complies with our editorial policies. Please see [our revision file checklist](https://www.nature.com/documents/CommsBio-file-checklist-revision.pdf) for guidance on formatting the manuscript and complying with our policies. You will also find guidelines for replying to the referees' comments. You may also wish to review our formatting guidelines for final submissions [here](https://www.nature.com/documents/commsj-life-style-formatting-guide-accept.pdf).

Please use the following link to submit your revised manuscript, point-by-point response to the referees' comments (which should be in a separate document to the cover letter) and any additional files:

<https://mts-commsbio.nature.com/cgi-bin/main.plex?el=A6Cx4DKg2B2GCB5I6A9ftdQ24GOjt0R81pjrFvVhvRwZ>

When submitting the revised version of your manuscript, please pay close attention to our [Digital Image Integrity Guidelines](https://www.nature.com/commsbio/editorial-policies/image-integrity) and to the following points below:

We would expect revisions of this nature to take around three months, but appreciate that every situation is unique. We look forward to receiving your revised manuscript when it is ready, and will not enforce a hard deadline on this revision.

Please do not hesitate to contact me if you have any questions or would like to discuss these

revisions further. We look forward to seeing the revised manuscript and thank you for the opportunity to review your work.

Best regards,

Eve Rogers, PhD
Associate Editor, Communications Biology
4 Crinan Street
London N1 9XW, UK
orcid.org/0000-0002-1841-7942
eve.rogers.1@nature.com

Reviewers' comments:

Reviewer #1 (Remarks to the Author):

I appreciate that the authors have considered our suggestions and made some statement and new figures, which improved the manuscript. However, there are still some problems.

1. What is the magnetic field intensity at the position of the mice bodies?
2. Were 5-mm thick copper plates used in mice study? The author also should provide a more detailed statement about animal experiment.
3. For the behavioral tests, which always have very big variations, the number of animals in this study is to low.

Reviewer #3 (Remarks to the Author):

- 1) Although many endpoints were detected such as mitochondrial superoxide, mitochondrial mass, and mitochondrial membrane potential, the association among these endpoints are not clear.
- 2) How many independent experiments have been carried out?
- 3) Were positive controls used in experiments to validate to expected outcome?
- 4) Which sensor were used to measure temperature? Given the distance between cells and coil, 300uT field should increase the temperature even though it's small amount.
- 5) Possible short term and long-term effects need to be discussed in discussion section. For example, what could be the reason of not seeing acute effects even though mitochondrial membrane potential in the mouse liver increased by ~@?
- 6) It's better to have both control and treated units in the same incubator for future experiments.

We cordially appreciate scrutinizing comments by the reviewers. Our specific responses are listed below and the revisions are highlighted in the revised manuscript, and are pasted in this letter.

Comments by Reviewer #1

Comment Rev1_1

What is the magnetic field intensity at the position of the mice bodies?

Response to Rev1_1

We appreciate the suggestion. We measured the magnetic field intensities in a mouse cage, and indicated them by color coding in Fig S1d. We added the following statements in Materials and Methods and Figure legend for Fig. S1.

Added statements in Materials and methods

Exposure of wild-type mice to ELF-WMF

All the studies on mice were approved by the Animal Care and Use Committee of Nagoya University, and were conducted in accordance with the relevant guidelines. Seven-week-old C57BL6/N male mice were purchased from Japan SLC. Two ELF-WMF devices were placed in tandem beneath the mouse cage (Fig. S1c). The intensities of magnetic fields, to which the mouse body was exposed, were from 5.3 to 14.2 μ T above the coil (Fig. S1d). Four to five mice were housed in a single cage for 4 weeks, and a total of 14 mice were analyzed for each of the ELF-WMF and control groups. For the control group, the cage was placed above the ELF-WMF device, but the switch was turned off. The Opti-ELF-WMF condition stated above was applied to the ELF-WMF group. To evaluate the effect of ELF-WMF in a conventional environment, we did not use 5-mm thick copper plates in the mouse study.

Revised legend for Supplementary Fig. S1

a. ELF-WMF setup for cultured cell. Four of the six wells were used for cell culture. All cells were fit within a 10.5-cm coil. Cells were sandwiched by copper plates, and the coil was placed 4 cm above the bottom plate. Current was applied clockwise when viewed from the top. Cells were cultured ~1 cm above the top of the coil, and the magnetic field intensities can be referred to the color coding in **d.** **b.** Electric current controller for the ELF-WMF coil. **c.** Two ELF-WMF coils were placed directly beneath a cage for mouse studies. Current was applied clockwise for both coils when viewed from the top. **d.** Color coding of measured magnetic field intensities in a mouse cage above a coil. Exposure to Opti-ELF-WMF for 4 weeks had no effect on fast (**e**) and slow (**f**) movements of wild-type mice as determined by an open-field locomotor activity test (mean \pm SD, $n = 14$ mice each; no statistical [n.s.] difference by Kruskal–Wallis test).

Comment Rev1_2

Were 5-mm thick copper plates used in mice study? The author also should provide a more detailed statement about animal experiment.

Response to Rev1_2

We appreciate the comment. We did not use 5-mm thick copper plates in our mouse study, because we hoped to analyze the effect of ELF-WMF in a conventional environment where the geomagnetic field exists. Mice were housed at the bottom of a breeding rack where an electromagnetic field generated by a ventilation motor at the top

of the rack could not be detected by our device (10 mG sensitivity). We added relevant statements in Materials and Methods, which was pasted above in Response to Rev1_1.

Comment Rev1_3

For the behavioral tests, which always have very big variations, the number of animals in this study is too low.

Response to Rev1_3

We appreciate the valuable suggestion. As suggested, we increased the number of mice from 4 to 14. We found that the standard deviation remained high, and the statistical difference remained nonsignificant. We updated the graphs in Supplementary Fig. S1ef, and revised a relevant legend, which was pasted above in Response to Rev1_1.

Comments by Reviewer #2

Comment Rev2_1

State direction of currents in two loops of exposure cage.

Response to Rev2_1

We appreciate the important suggestion. We always applied currents clockwise when viewed from the top. We added relevant statements in legends for Fig S1ac, which was pasted above in Response to Rev1_1.

Comment Rev2_2

Add mitochondria specific VDAC1 to Figure 4c.

Response to Rev2_2

As suggested, we performed immunoblotting of VDAC1 in whole cell extracts and mitochondrial fractions in Figure 4b.

Comment Rev2_3

Measure mitochondrial mass and mitochondrial membrane potential at 3 h and 12 h

Response to Rev2_3

Thank you for the suggestion. We showed mitochondrial mass and mitochondrial membrane potential at 3 h and 12 h in Fig. 2bc. Statistical significance was observed in the reduction of mitochondrial mass at 3 h (Fig. 2b) and in the increase of mitochondrial membrane potential at 12 h (Fig. 2c). We clearly indicated them in Results, and summarized our findings at the beginning of Discussion.

Revised statement in Results

Opti-ELF-WMF most decreased the level of mitochondrial superoxide at 1 h, mitochondrial mass at 3 h, and mitochondrial membrane potential at 6 h, and most increased them at 12 h (Fig. 2a, b, c).

Revised statement in Discussion

We found that Opti-ELF-WMF reduced the amount of mitochondria by ~30% (Fig. 2b) by inhibiting mitochondrial ETC complex II by ~15% (Fig. 6a), which subsequently induced mitophagy (Fig. 4) and increased mitochondrial membrane potential (Fig. 2c).

Comments by Reviewer #3

Comment Rev3_1

Although many endpoints were detected such as mitochondrial superoxide, mitochondrial mass, and mitochondrial membrane potential, the association among these endpoints are not clear.

Response to Rev3_1

Thank you for the precious suggestion. We started our discussion by summarizing what we found and by citing relevant figures.

Revised first statement in Discussion

We found that Opti-ELF-WMF reduced the amount of mitochondria by ~30% (Fig. 2b) by inhibiting mitochondrial ETC complex II by ~15% (Fig. 6a). This subsequently induced mitophagy (Fig. 4) to eliminate damaged mitochondria, and later activated mitochondrial biogenesis (Fig. 5) to increase mitochondrial membrane potential (Fig. 2c).

Comment Rev3_2

How many independent experiments have been carried out?

Response to Rev3_2

We confirmed that the number of experiments was indicated throughout our manuscript (highlighted). For example, “ $n = 3$ mice” indicates three independent mice were used for the study, but the mice were purchased from the same company and some of them might be siblings. Similarly, “ $n = 3$ culture dishes” indicates cells were cultured in three independent dishes, but the cells were split from an identical batch of cultured cells.

Comment Rev3_3

Were positive controls used in experiments to validate to expected outcome?

Response to Rev3_3

We cordially appreciate the valuable suggestion. We assume that positive controls are required when we observe no effect. The only experiment that we observed no effects was the assay of fast and slow movements of mice exposed to Opti-ELF-WMF for 4 weeks in Fig. S1de. We could not find any stimulus that serves as a positive control in this experiment. Instead, we increased the number of mice of this experiment from 4 to 14, and found that the statistical significance remained unobserved with 14 mice each.

Comment Rev3_4

Which sensor were used to measure temperature? Given the distance between cells and coil, 300 uT field should increase the temperature even though it's small amount.

Response to Rev3_4

Thank you for the suggestion. We indicated that we used SN3000 by Netsuken that had a resolution of 0.1°C in Materials and Methods. We continuously measured the temperature of culture medium in a 10-cm cultured dish that was placed above the coil in a 37°C incubator for 24 h. We observed that the temperature was not changed even when the ELF-WMF stimulus of 300 μ T was applied.

Added statement in Materials and Methods

We confirmed that the application of the ELF-WMF stimulus of 300 μ T for 24 h did not change the temperature of the culture medium in a 10-cm culture dish placed above the coil by a digital thermometer at 0.1°C resolution (SN3000, Netsuken).

Comment Rev3_5

Possible short term and long-term effects need to be discussed in discussion section. For example, what could be the reason of not seeing acute effects even though mitochondrial membrane potential in the mouse liver increased by ~40%?

Response to Rev3_5

Thank you for your precious suggestion. We observed the effects of Opti-ELF-WMF on the mouse liver to examine whether the long-term effects can be observed or not. We revised relevant statements in Discussion.

Revised statements in Discussion

We found that Opti-ELF-WMF reduced the amount of mitochondria by ~30% (Fig. 2b) by inhibiting mitochondrial ETC complex II by ~15% (Fig. 6a). This subsequently induced mitophagy (Fig. 4) to eliminate damaged mitochondria, and later activated mitochondrial biogenesis (Fig. 5) to increase mitochondrial membrane potential (Fig. 2c). To examine the long-term effects of Opti-ELF-WMF, we exposed wild-type mice to Opti-ELF-WMF for 4 weeks, and observed increased mitochondrial membrane potential in the mouse liver by ~40% (Fig. 1b).

Comment Rev3_6

It's better to have both control and treated units in the same incubator for future experiments.

Response to Rev3_6

We appreciate productive and encouraging suggestions. We totally agree with you. A nickel-iron magnetic alloy, permalloy, would shield the electromagnetic fields in our incubator. However, we found that the price of permalloy was too much expensive for us. We hope that we can afford permalloy in our future projects.

REVIEWERS' COMMENTS:

Reviewer #1 (Remarks to the Author):

The authors have revised the manuscript according to the suggestions. This manuscript reported that the effects of very weak magnetic fields could reduce mitochondrial mass to 70% and electron transport chain complex II activity in liver, which induces mitophagy and rejuvenates mitochondria. Although more experiments are needed to fully support this conclusion, the authors have provided a starting point for further research in this direction in the future.

Reviewer #4 (Remarks to the Author):

There are many studies looking for magnetic field effects weaker than earth's magnetic field in ELF. Modifying the following sentence would be helpful: "The in vitro or in vivo effects of ELF-WMF, weaker than the geomagnetic field, have not been reported to the best of our knowledge."

Complex I is the largest enzyme of mitochondria. Why inhibiting Complex II is important and how it's relevant to other complexes? Authors mention that Complex II has iron Sulphur clusters. However, Complex I have the most Iron Sulphur clusters. A clear distinction needs to be made.

Comment by Reviewer #1

Comment Rev1_1

The authors have revised the manuscript according to the suggestions. This manuscript reported that the effects of very weak magnetic fields could reduce mitochondrial mass to 70% and electron transport chain complex II activity in liver, which induces mitophagy and rejuvenates mitochondria. Although more experiments are needed to fully support this conclusion, the authors have provided a starting point for further research in this direction in the future.

Response to Rev1_1

We cordially appreciate encouraging comments. We will further work on the underlying mechanisms.

Comment by Reviewer #4

Comment Rev4_1

There are many studies looking for magnetic field effects weaker than earth's magnetic field in ELF. Modifying the following sentence would be helpful: "The in vitro or in vivo effects of ELF-WMF, weaker than the geomagnetic field, have not been reported to the best of our knowledge."

Response to Rev4_1

We apologize for ignorance of some important studies showing the biological effects of extremely weak magnetic fields. We scrutinized PubMed database again and found relevant articles. We revised our statement as follows.

Revised statement in Introduction

The biological effects of ELF-WMF, weaker than the geomagnetic field, have been reported in cultured cells⁷, planaria⁸, rats⁹, lizards^{10,11}, and humans¹², but the underlying mechanisms remain mostly elusive.

Comment Rev4_2

Complex I is the largest enzyme of mitochondria. Why inhibiting Complex II is important and how it's relevant to other complexes? Authors mention that Complex II has iron Sulphur clusters. However, Complex I have the most Iron Sulphur clusters. A clear distinction needs to be made.

Response to Rev4_2

Thank you for the suggestion. We agree that the iron sulfur cluster is not unique to complex II. We revised our statement in Discussion as follows.

Added statement in Discussion

The mitochondrial ETC complex II and Cry/MagR complex share the same components: flavin adenine dinucleotide (FAD) and iron-sulfur clusters. As mitochondrial ETC complexes I, II, and III carry 8, 3, and 1 iron-sulfur clusters, FAD alone or a combination of FAD and iron-sulfur clusters may account for the effects of ELF-WMF. A moiety in the mitochondrial ETC complex II that is targeted by Opti-ELF-WMF may reside in a structure shared with the Cry/MagR complex.